# Organ geometry channels reproductive cell fate in the Arabidopsis ovule primordium

Elvira Hernandez-Lagana[1†], Gabriella Mosca[2†], Ethel Mendocilla-Sato[2†], Nuno Pires[2], Anja Frey[2‡], Alejandro Giraldo-Fonseca[2], Caroline Michaud[1], Ueli Grossniklaus[2], Olivier Hamant[3], Christophe Godin[3], Arezki Boudaoud[3§], Daniel Grimanelli[1], Daphné Autran[1,3]*, Célia Baroux[2]*

[1]DIADE, University of Montpellier, CIRAD, IRD, Montpellier, France; [2]Department of Plant and Microbial Biology and Zurich-Basel Plant Science Center, University of Zürich, Zürich, Switzerland; [3]Laboratoire Reproduction et Développement des Plantes, University of Lyon, ENS Lyon, UCB Lyon 1, CNRS, INRAE, INRIA, Lyon, France

*For correspondence:
daphne.autran@ird.fr (DA);
cbaroux@botinst.uzh.ch (CB)

[†]These authors contributed equally to this work

Present address: [‡]Novartis Pharma Schweiz AG, Basel, Switzerland; [§]LadHyX, Ecole polytechnique CNRS, Institut Polytechnique de Paris, Palaiseau, France

**Abstract** In multicellular organisms, sexual reproduction requires the separation of the germline from the soma. In flowering plants, the female germline precursor differentiates as a single spore mother cell (SMC) as the ovule primordium forms. Here, we explored how organ growth contributes to SMC differentiation. We generated 92 annotated 3D images at cellular resolution in Arabidopsis. We identified the spatio-temporal pattern of cell division that acts in a domain-specific manner as the primordium forms. Tissue growth models uncovered plausible morphogenetic principles involving a spatially confined growth signal, differential mechanical properties, and cell growth anisotropy. Our analysis revealed that SMC characteristics first arise in more than one cell but SMC fate becomes progressively restricted to a single cell during organ growth. Altered primordium geometry coincided with a delay in the fate restriction process in *katanin* mutants. Altogether, our study suggests that tissue geometry channels reproductive cell fate in the Arabidopsis ovule primordium.

## Introduction

A hallmark of sexual reproduction in multicellular organisms is the separation of the germline from the soma. In animals, primordial germ cells (PGCs) are set-aside during embryogenesis from a mass of pluripotent cells. The number of germ cells depends on the balance between proliferation (self-renewal) and differentiation, a process controlled by both intrinsic factors and signals from the surrounding somatic tissues. In flowering plants, the first cells representing the germline, the spore mother cells (SMCs), differentiate only late in development. SMCs arise multiple times, in each flower during the formation of the reproductive organs. In Arabidopsis, the female SMC differentiates in the nucellus of the ovule primordium, a digit-shaped organ that emerges from the placental tissue of the gynoecium. The SMC is recognizable as a single, large, and elongated subepidermal cell, which is centrally positioned within the nucellus and displays a prominent nucleus and nucleolus (*Bajon et al., 1999*; *Bowman, 1993*; *Schmidt et al., 2015*; *Schneitz et al., 1995*).

Although SMC singleness may appear to be robust, more than one SMC candidate *per* primordium is occasionally seen, yet at different frequencies depending on the specific *Arabidopsis* accession (~5% in Landsberg *erecta* [L*er*], 10% in Columbia [Col-0], 27% in Monterrosso [Mr-0]), (*Grossniklaus and Schneitz, 1998*; *Rodríguez-Leal et al., 2015*). *Arabidopsis*, maize, and rice mutants in which SMC singleness is compromised have unveiled the role of regulatory pathways

involving intercellular signaling, small RNAs, as well as DNA and histone methylation (*Garcia-Aguilar et al., 2010*; *Mendes et al., 2020*; *Nonomura et al., 2003*; *Olmedo-Monfil et al., 2010*; *Schmidt et al., 2011*; *Sheridan et al., 1996*; *Sheridan et al., 1999*; *Su et al., 2020*; *Su et al., 2017*; *Zhao et al., 2008*). As the SMC forms, cell-cycle regulation contributes to the stabilization of its fate in a cell-autonomous manner through cyclin-dependent kinase (CDK) inhibitors and RETINOBLAS-TOMA-RELATED1 (RBR1) (*Cao et al., 2018*; *Zhao et al., 2017*). SMC singleness thus appears to result from a two-step control: first, by restricting differentiation to one cell and second, by preventing self-renewal before meiosis (reviewed in *Lora et al., 2019*; *Pinto et al., 2019*).

However, the precise mechanisms underlying the plasticity in the number of SMC candidates and SMC specification are still poorly understood. In principle, SMC singleness may be controlled by successive molecular cues. However, even in that scenario, such cues must be positional, at least to some extent, and thus involve a spatial component. Over the last decade, many different molecular cues defining spatial patterns in the ovule primordium were identified (*Pinto et al., 2019*; *Su et al., 2020*); however, their coordination is unknown. Since SMCs emerge at the primordium apex concomitant with its elongation, we hypothesize that geometric constraints during ovule morphogenesis influence SMC singleness and differentiation. Such a hypothesis could explain variation in the number of SMC candidates, ultimately culminating in a single SMC entering meiosis. Answering the questions of whether SMC formation follows a stereotypical or plastic developmental process and whether it is intrinsically linked to or independent of ovule primordium formation would unravel fundamental principles connecting cell fate establishment and organ growth.

Such an analysis requires a high-resolution description of ovule geometry during development. Our current knowledge of ovule primordium growth in *Arabidopsis* is based on two-dimensional (2D) micrographs from tissue sections or clearing. It is described in discrete developmental stages capturing classes of primordia by their global shape and SMC appearance until meiosis and by the presence of integument layers and ovule curvature later on (*Grossniklaus and Schneitz, 1998*). In addition, a 3D analysis of average cell volumes during primordium growth was recently provided (*Lora et al., 2017*), and extensive 3D analysis was carried on for late ovule stages (*Vijayan et al., 2021*). Yet, we lack a view of the patterning processes regulating early ovule primordium formation and how the dynamics of cell proliferation contributes to the cellular organization during primordium growth. Thus, we described and quantified the growth of the ovule primordium at cellular resolution in 3D. We combined 3D imaging, quantitative analysis of cell and tissue characteristics, reporter gene analyses, and 2D mechanical growth simulations. In addition, using the *katanin* mutant that affects anisotropic cell growth and division patterns (*Luptovčiak et al., 2017a*; *Ovečka et al., 2020*), we show that altered ovule morphology leads to ectopic SMC candidates. We also uncovered that differentiation of SMC candidates initiate earlier than previously thought, and provide evidence for a gradual process of cell fate restriction, channeling the specification of a single SMC prior to meiosis.

## Results

### Building a reference image dataset capturing ovule primordium development at cellular resolution

To generate a reference image dataset describing ovule primordium development in 3D and with cellular resolution, we imaged primordia at consecutive stages in intact carpels by confocal microscopy. Carpels were cleared and stained for cell boundaries using a modified PS-PI staining (*Truernit et al., 2008*) and mounted using a procedure preserving their 3D integrity (*Mendocilla Sato and Baroux, 2017*; *Figure 1A*). We selected high signal-to-noise ratio images and segmented them based on cell boundary signals using Imaris (Bitplane, Switzerland) as described previously (*Mendocilla Sato and Baroux, 2017*; *Figure 1B*). We manually curated 92 ovules representing seven consecutive developmental stages (7–21 ovules *per* stage, *Figure 1B*, *Table 1*, *Figure 1—source data 1*) and classified them according to an extended nomenclature (explained in Materials and methods). The temporal resolution of our analysis led us to subdivide early stages (stage 0-I to stage 0-III) covering primordium emergence prior to the straight digit-shape of the organ set as stage 1-I, where the SMC becomes distinguishable by its apparent larger size in longitudinal views (*Grossniklaus and Schneitz, 1998*; *Figure 1B*).

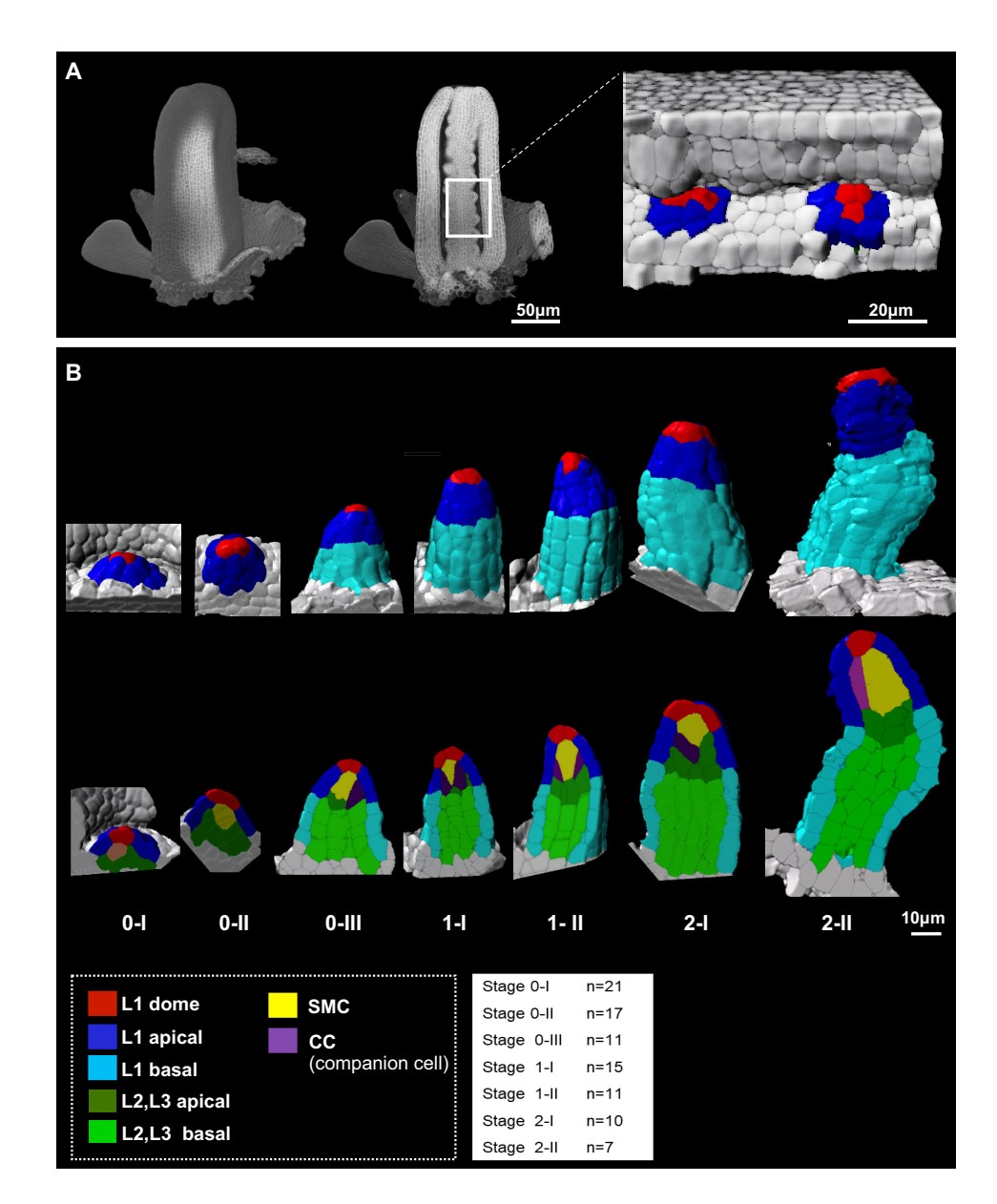

**Figure 1.** Reference set of 3D segmented images capturing Arabidopsis ovule primordium growth at cellular resolution. (**A**) 3D reconstruction of a whole gynoecium stained with PS-PI (cell wall dye) and visualized by CSLM. The cross-section shows nascent ovule primordia attached to the placenta. (**B**) Ovule primordium developmental stages (0-I to 2-II) and organ viewpoints (domains) defined for 3D quantitative analyses. All segmented data can be analyzed by an interactive interface named OvuleViz. See also *Figure 1—figure supplement 1A–C*. n = number of ovules analyzed. Segmented images for all developmental stages are provided in *Figure 1—source data 1*. See also *Figure 1—figure supplement 1D–E*, Materials and methods.

The online version of this article includes the following source data and figure supplement(s) for figure 1:

**Source data 1.** Image gallery.

**Figure supplement 1.** Approaches for morphodynamic analyses of ovule primordium development using a reference set of 3D segmented images.

To evaluate the distinct contribution of domain-, layer-, and cell-specific growth dynamics, we labeled cells depending on their location in different regions of the ovule primordium: apical vs basal

**Table 1.** Classification criteria of Arabidopsis ovule primordia.

The table summarizes general characteristics of ovule primordia per stage: cell 'layers' above the placenta scored as the number of L1 cells in a cell file drawn from the basis to the top, range thereof, total cell number, ovule shape including height, width, aspect ratio. See also *Figure 1—source data 1*.

| | 0-I | | | 0-II | | | 0-III | | |
|---|---|---|---|---|---|---|---|---|---|
| Cells above placenta* | 2.5 | (±3.5) | (n = 21) | 3.6 | (±0.5) | (n = 17) | 5.9 | (±1.0) | (n = 11) |
| Range (min-max) | 2 | - | 3 | 3 | - | 4 | 5 | - | 8 |
| Total # cells | 28 | (±5.7) | (n = 21) | 38 | (±8.0) | (n = 17) | 80 | (±8.4) | (n = 11) |
| Width (µm) (W) | 23 | (±3.5) | | 26.3 | (±2.4) | | 28.0 | (±3.1) | |
| Height (µm) (H) | 5.3 | (±1.2) | | 11.9 | (±2.1) | | 22.5 | (±3.6) | |
| H:W ratio | 0.2 | (±0.04) | (n = 13) | 0.45 | (±0.06) | (n = 14) | 0.8 | (±0.12) | (n = 8) |

| | 1-I | | | 1-II | | |
|---|---|---|---|---|---|---|
| Cells above placenta* | 7.3 | (±1.3) | (n = 15) | 9.7 | (±1.0) | (n = 11) |
| Range (min-max) | 6 | - | 10 | 8 | - | 11 |
| Total # cells | 105 | (±10.6) | (n = 15) | 128 | (±18.0) | (n = 11) |
| Width (µm) (W) | 27 | (±2.0) | | 27.8 | (±3.5) | |
| Height (µm) (H) | 30 | (±2.8) | | 39 | (±5.9) | |
| H:W ratio | 1.1 | (±0.10) | (n = 6) | 1.4 | (±0.22) | (n = 8) |

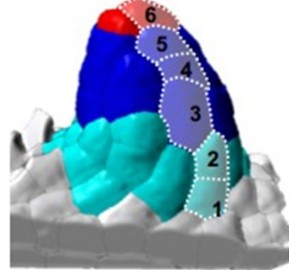

* scoring the length of L1 cell file above placenta

| | 2-I | | | 2-II | | |
|---|---|---|---|---|---|---|
| Cells above placenta* | 10 | (±1.1) | (n = 10) | 12.6 | (±1.0) | (n = 7) |
| Range (min-max) | 9 | - | 12 | 11 | - | 14 |
| Total # cells | 151 | (±18.9) | (n = 10) | 165 | (±19.8) | (n = 7) |

domains, L1,L2,L3 layers. In addition, we associated each cell with a cell type: 'L1 apical', 'L1 basal', 'L2,L3 apical', 'L2,L3 basal', 'SMC', 'L1 dome' (for the upmost apical L1 cells in contact with the SMC), 'CC' (for companion cells, elongated L2 cells adjacent to the SMC) (*Figure 1B*, Materials and methods).

To generate a quantitative description of ovule primordia with respect to cell number, size, and shape according to cell labels, layers, domains, ovule stage, and genotype, we developed an interactive, R-based interface named OvuleViz. The interface imports cell descriptors exported from segmented image files and enables multiple plots from a user-based selection of (sub)datasets (*Figure 1—figure supplement 1A–C*, Materials and methods). This work generated a reference collection of annotated, 3D images capturing ovule primordium development at cellular resolution from emergence until the onset of meiosis. The collection of 92 segmented images, comprising a total of 7763 annotated cells and five morphological cell descriptors (volume, area, sphericity, prolate and oblate ellipticity), provides a unique resource for morphodynamic analyses of ovule primordium growth.

To identify correlations between growth patterns and differentiation, we first performed a principal component analysis (PCA) based on the aforementioned cell descriptors, *per* cell type and stage, considered together or separately (*Figure 1—figure supplement 1D–E*). In this global analysis, the SMC appears morphologically distinct at late stages (2-I and 2-II) but not at early stages. This prompted us to investigate in detail the contribution of different layers, domains, and cell types to ovule primordium growth and in relation to SMC differentiation.

## Ovule primordium morphogenesis involves domain-specific cell proliferation and anisotropic cell shape patterns

The ovule primordium emerges from the placenta as a small dome-shaped protrusion and grows into a digit-shaped primordium with nearly cylindrical symmetry (stage 1-I) before enlarging at the base (*Figure 1B*). Using our segmented images, we first quantified global changes in cell number, cell volume, and ovule primordium shape. Our analysis revealed two distinct phases of

morphological events. Phase I (stages 0-I to 0-III) is characterized by a 4.5-fold increase in total cell number together with a moderate increase in mean cell volume (10%, p=0.03). By contrast, Phase II (stages 1-I to 2-II) is characterized by a moderate increase in cell number (50%) and the global mean cell volume is relatively constant (*Figure 2A*, *Figure 2—figure supplement 1A*). To quantify the resulting changes in organ shape, we extrapolated a continuous surface mesh of the ovule outline and used it to compute its height and width at the base (*Figure 2B–C*, *Figure 2—figure supplement 1B*, Appendix 1). Anisotropic organ growth during Phase I was confirmed by a steady increase in height, while primordium width increased moderately (*Figure 2C*). This contrast in events between Phase I and II is illustrated by the fold-changes (FCs) in cell number and aspect ratio (*Figure 2D*), which range between 1.5 and 2.0 in Phase I but drop to 1.4 and 1.2 in Phase II. These observations confirmed that Phase I shows distinct growth dynamics compared to Phase II.

Next, to capture possible specific patterns of growth, we analyzed cell number, cell size, and cell shape using different viewpoints: one comparing the L1 and L2-L3 layers and one contrasting the apical *vs.* basal domains. Counting cell number *per* viewpoint clearly showed a dominant contribution of the epidermis (L1) relative to the subepidermal layers and of the basal relative to the apical domain (*Figure 2E and F*). To verify these findings with a cellular marker, we analyzed the M-phase-specific *promCYCB1.1::CYCB1.1-db-GFP* reporter (abbreviated CYCB1.1db-GFP) (*Ubeda-Tomás et al., 2009*). We scored the number of GFP-expressing cells among 481 ovules and plotted relative mitotic frequencies *per* cell layer and domain for each ovule stage to generate a cell-based mitotic activity map (*Figure 2G–H*, *Figure 2—source data 1*, Materials and methods). In this approach, subepidermal (L2) cells beneath the dome were distinguished from underlying L3 cells to gain resolution in the L2 apical domain where the SMC differentiates. Consistent with our previous observation, in Phase I, a high proliferation activity was scored in L1 cells at the primordium apex (scoring 64% of all mitotic events). By contrast, the L2 apical domain remains relatively quiescent (only 3% of the mitotic events). During Phase II, the majority (60%) of mitotic events is found in the basal domain, consistent with the progressive population of the basal domain with more cells. It is of note that during this phase, few mitotic events are detected in L2 apical cells, with the exception of SMC neighbor cells that show frequent divisions at stage 1-II. Thus, reporter analysis confirmed a biphasic, temporal pattern of cell division with changing regional contributions, suggesting the L1 dome and the basal domain as consecutive sites of proliferation, contributing to the morphological changes in Phase I and II, respectively.

Average cell size analysis, by contrast, did not reveal significant changes during primordium development with the notable exception of the SMC (*Figure 2A*, *Figure 2—figure supplement 1C*). The distinct size of the SMC candidate is already detected at stage 0-III when compared to other L2, L3 cells (*Figure 2I*), or even earlier (stage 0-I) when compared to all other cells (*Figure 2I* inset). Size differentiation of the SMC is not uniform among ovules, demonstrating plasticity in the process (*Figure 2—figure supplement 1D*). In addition, cells from the L2 and L3 layers are larger than L1 cells already at stage 0-I (*Figure 2J*, *Figure 2—figure supplement 1E*), possibly due to a longer growth phase, consistent with the low division frequency observed previously.

We then investigated cell shape changes during primordium elongation, using ellipticity and sphericity indices computed following segmentation. The analysis did not reveal significant differences between domains or layers (*Figure 2—figure supplement 1F*). This could indicate either a highly variable cell shape or local, cell-specific differences. We thus more specifically analyzed the subepidermal domain where the SMC differentiates. Companion cells showed an increasing ellipticity (and decreasing sphericity), starting at stage 1-I and culminating at stage 2-I (*Figure 2—figure supplement 1F*). By contrast, the SMC only showed a moderate decrease in sphericity at late stages and no distinctive ellipticity at early stages (*Figure 2—figure supplement 1F*) when compared to other cells. To get more information on SMC shape, we compared the maximum, medium, and minimum anisotropy index. For this, we developed an extension for MorphoMechanX (*Barbier de Reuille et al., 2015*) (http://www.morphomechanx.org), (i) to perform a semi-automatic labeling of cell layers and cell types from a cellularized mesh obtained from segmentation data, and (ii) to compute in each 3D cell the principal axes of shape anisotropy and the corresponding indices (*Figure 2—figure supplement 1G*, Appendix 1, *Figure 2—video 1*). The averaged maximum anisotropy shape index of the SMCs was consistently above the medium (and hence above the minimum) anisotropy index (*Figure 2K*, *Figure 2—figure supplement 1H*). We measured the degree of alignment of the SMC major axis with the main growth axis of the ovule during early stages (stage 0-II) and found a

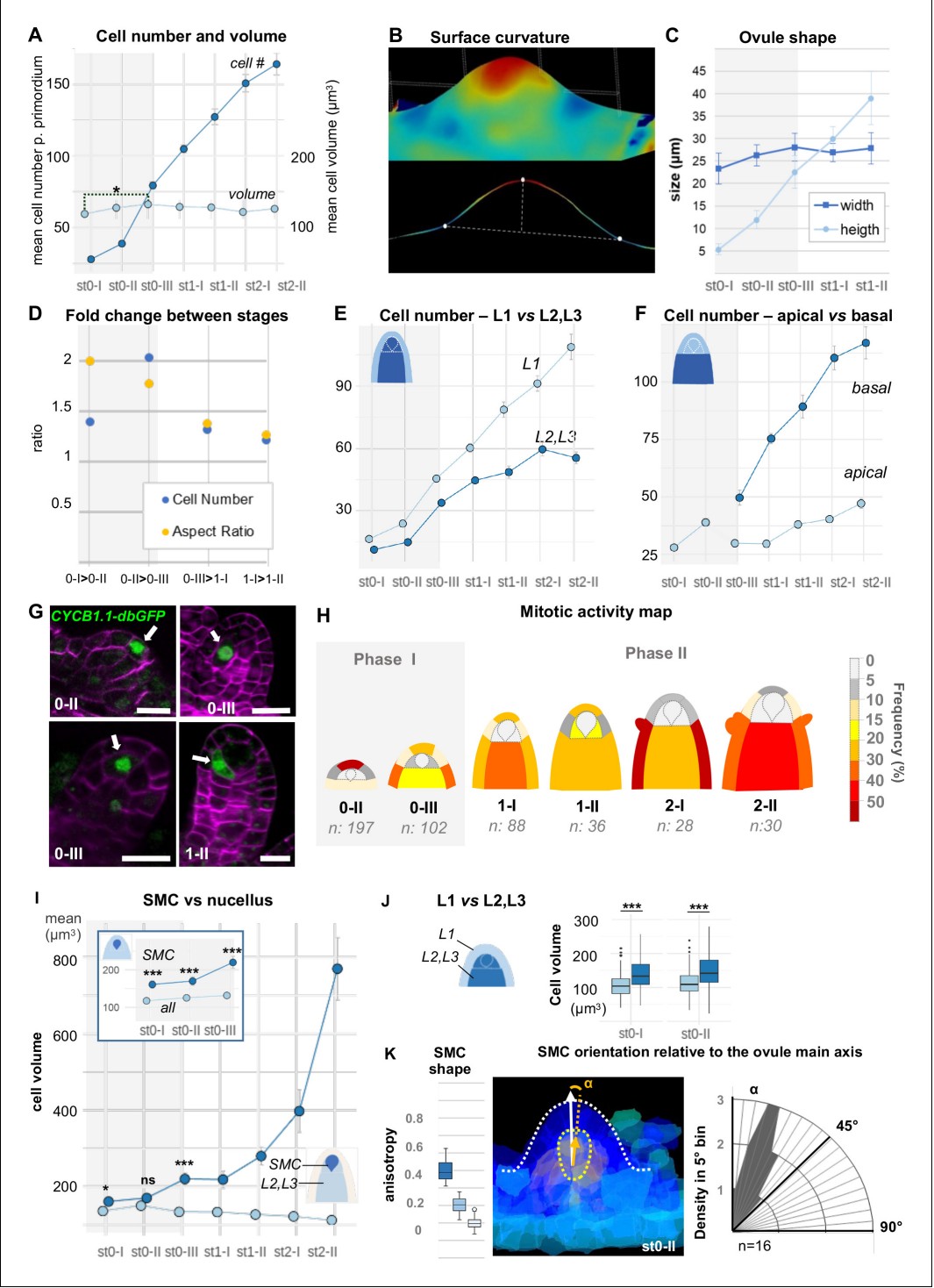

**Figure 2.** Ovule primordium morphogenesis involves domain-specific cell division and anisotropic cell growth. (**A**) Mean cell number *per* ovule increases mainly during stages 0-I to 0-III (Phase I), whereas cell volume *per* ovule remains constant on average across primordium development (stages 0-I to 2-II). (**B**) Representative image of a continuous surface of an ovule primordium mesh and its projected median plane. Dashed lines indicate the minimal and maximal curvature points used to measure organ height and width. Color scale: minimal curvature mm-1 (see also ***Figure 2—figure supplement 1B***). (**C**) Anisotropic organ growth during Phase I and until stage 1-II. Mean width and height were quantified *per* stage. (**D**) Phase I shows distinct growth dynamics compared to Phase II. Fold-change of cell number and aspect ratio between stages are plotted. (**E**) Mean cell number is increased at the L1 *vs.* L2,L3 layers across developmental stages. (**F**) Mean cell number is increased in the basal *vs.*

*Figure 2 continued on next page*

*Figure 2 continued*

apical domain across developmental stages. (G) Representative images of ovule primordia expressing the M-phase reporter *promCYCB1.1::CYCB1.1db-GFP* (CYCB1.1db-GFP). White arrows indicate dividing cells. Magenta signal: Renaissance SR2200 cell-wall label. Scale bar: 10 μm. (H) Domain-specific map of mitotic activity during ovule primordium development, scored using the CYCB1.1db-GFP reporter. The frequency of mitoses was calculated *per* ovule domain at each developmental stage and color-coded as indicated in the bar (right). n: total number of scored ovules. (I) Mean SMC candidate volume (dark blue) is significantly increased as compared to L2, L3 cells (pale blue) from stage 0-III onward, on even at earlier stages as compared to all other cells (inset). (J) Mean cell volume is increased in L2,L3 cells as compared to L1 cells, at the two early developmental stages (0-I, 0-II). (K) The SMC consistently displays anisotropic shape with main axis of elongation aligned with ovule growth axis. The SMC anisotropy index (boxplot, left; stage 0-II, n = 16 ovules) was calculated from the Maximum (dark blue), Minimum (light blue), and Medium (medium blue) covariance matrix eigenvalues, computed from 3D segmented cells (see *Figure 2—figure supplement 1G*). Image (middle): illustration of the SMC main anisotropy axis (orange arrow) related by an angle 'alpha' to the main axis of the ovule primordium (white arrow), stage 0-II. Radar plot (right): 'alpha' angle measured on z projections for n = 16 ovule primordia at stage 0-II. See also Materials and methods. Error bars: Standard errors to the mean. Differences between cell types or primordium domains were assessed using a two-tailed Man Whitney U test in (A) and (I); a two-tailed Wilcoxon signed rank test in (J). *p≤0.05, **p≤0.01, ***p≤0.001. See also *Figure 2—figure supplement 1*, *Figure 2—source data 1*, Materials and methods.

The online version of this article includes the following video, source data, and figure supplement(s) for figure 2:

**Source data 1.** Raw data for quantitative analysis.

**Figure supplement 1.** Ovule primordium morphogenesis involves domain-specific cell division and anisotropic cell growth patterns.

**Figure 2—video 1.** Semi-automated cell-type classification and anisotropy quantification with MorphoMechanX.
https://elifesciences.org/articles/66031#fig2video1

mean angle of 22° (±11°, n = 16) (*Figure 2K*). This confirmed that the SMC has a consistent anisotropic shape from early stages onwards, with a distinguishable major axis aligned with the primordium axis.

Taken together, these results suggest that anisotropic primordium growth is linked to a biphasic, domain-specific cell proliferation pattern, alternating between the L1 dome at Phase I and the basal domain at Phase II, combined with localized, anisotropic expansion in the L2 apical domain. In this process, SMC characteristics, such as distinct size, anisotropic shape, and orientation aligned with the growth axis of the primordium, emerge already in Phase I. The pronounced growth and elongation of the SMC in Phase II then occurs concomitant with primordium elongation. While primordium elongation is not explained by anisotropic cell growth alone but also depends on cell proliferation as shown above, the observation that cells are elliptic suggests a potential role for anisotropic cell growth. We explore this property in the next section through an in silico approach.

## 2D mechanical simulations relate ovule primordium growth to SMC shape emergence

Organ shape is determined by the rate and direction of cell growth, which is affected by signaling and the mechanical state and geometry of the tissue. This provides room for multiple regulatory feedback mechanisms and interactions between them (*Bassel et al., 2014*; *Echevin et al., 2019*). Mechanical constraints arise from the growth process in form of tensile and compressive forces that, in turn, influence cell and tissue growth (*Echevin et al., 2019*). To determine the role of ovule primordium growth on SMC differentiation, we sought to understand the contributions of local growth rate and anisotropy, and their relation to signaling and mechanical constraints (*Coen et al., 2004*; *Kennaway et al., 2011*).

For this, we developed two complementary 2D models of a longitudinal section of the ovule intersecting its main elongation axis: (i) a finite element method FEM-based mechanical model of the ovule represented as a continuous object (only outlining L1 *vs.* inner L2,L3 tissue) and (ii) a mass spring MS-based model of the ovule able to represent the mechanical status of each cell wall.

While MS-based models allow the investigation of the connection between organ and individual cell growth, the FEM-based models allow testing the role of material anisotropy. In addition, the two methods represent the cell wall in complementary ways. In FEM models, the cell wall is a

continuous material throughout the tissue representation, while in MS models, the cell wall is modeled as a network of connected elasto-plastic wires. The two approaches together allow us to determine the most plausible morphogenetic principles of ovule primordium growth, while addressing the contribution of specific parameters, such as material and cell-based properties in FEM and MS models, respectively.

Growth was implemented in both frameworks using two uncoupled but complementary modalities: (i) growth in which an abstract growth factor captures the cumulative effects of biochemical signals without considering their explicit mode of action (i.e. cell wall loosening, increasing turgor pressure), hereafter referred to as 'signal-based growth' (*Boudon et al., 2015*; *Coen et al., 2004*); and (ii) passive growth through relaxation of excess of strain inside the tissue, hereafter referred as 'strain-based growth' (*Boudon et al., 2015*; *Bozorg et al., 2016*).

The mechanical equilibrium is computed to ensure compatibility within the tissue that is locally growing at different rates and orientations. As a consequence, residual internal compressions and tensions arise and this, in turn, affects the mechanical behavior (*Boudon et al., 2015*; *Rodriguez et al., 1994*). A polarization field is used to set the direction of anisotropic growth (*Coen et al., 2004*). The simulation consists of cyclic iterations where the mechanical equilibrium is computed before each growth step is specified by signal-based growth or strain-based growth (Appendix 1). This strategy extends previous tissue growth models (*Bassel et al., 2014*; *Boudon et al., 2015*; *Bozorg et al., 2016*; *Kuchen et al., 2012*; *Mosca et al., 2018*). In the case of MS models, a cell will keep growing until it reaches a pre-assigned, user-defined target area, after which it will divide (shortest wall through the centroid rule, see Appendix 1, *Figure 3—figure supplement 1D*).

We designed a starting template consisting of an L1 layer distinct from the underlying L2,L3 tissue based on different growth and material properties (*Table 2*, *Figure 3—figure supplement 1*, Appendix 1). We first set the model components to produce a realistic, elongated, digit-shaped primordium with a narrow dome and an L1 layer of stable thickness during development, fitting experimental observations (*Figure 2—figure supplement 1C*) (Reference Model, FEM-Model 1, MS-Model 2) (*Figure 3A, C and E*, *Figure 3—figure supplement 1A–D*). This model combines the following hypotheses: an initial, narrow domain of anisotropic, signal-based growth with a high concentration in inner layers, a broad domain competent for passive strain-based relaxation, and material

**Table 2.** Hypotheses used to generate the mass-spring (MS)- and continuous Finite Element Method (FEM)-based simulations.
Several growth and mechanical hypotheses were listed at start of modeling. To evaluate their effect on primordium growth, each hypothesis was excluded (-) in at least one scenario. The FEM- and MS-based models presented in *Figure 3* and *Figure 3—figure supplement 1* are numbered according to scenarios 1–8 in the table. Since it is not possible for MS models to simulate material anisotropy, Model 1 was only tested with FEM models. The hypothesis of growth anisotropy is always active for the L1 layer in the models as reported in the table. Empty dots for 'material anisotropy' were considered only for the FEM models. See also *Figure 3—figure supplement 1* and Appendix 1 for modeling principles, results, and detailed computational methods.

| Modeling hypotheses | | Models | | | | | | | |
|---|---|---|---|---|---|---|---|---|---|
| | | 1 | 2 | 3 | 4 | 5 | 6 | 7 | 8 |
| Growth anisotropy | | ● | ● | - | ● | ● | ● | ● | ● |
| Material anisotropy | | ● | - | ○ | ○ | ○ | ○ | ○ | ○ |
| Strain-based growth | | ● | ● | ● | ● | ● | ● | ● | - |
| Signal-based growth | | | | | | | | | |
| Distribution | L1 only | - | - | - | - | - | ● | - | - |
| | Inner L2,L3 tissue only (pit-shape) | - | - | - | - | - | - | ● | - |
| | L1 + inner L2,L3 tissue (pit-shape) | ● | ● | ● | - | ● | - | - | ● |
| | L1 + inner L2,L3 tissue (broad distribution) | - | - | - | ● | - | - | - | - |
| Fixed high concentration | L1 only | - | - | - | - | ● | ● | - | - |
| | Inner L2,L3 tissue only | - | - | - | - | - | - | ● | - |
| | L1 + inner L2,L3 tissue | ● | ● | ● | ● | - | - | - | ● |

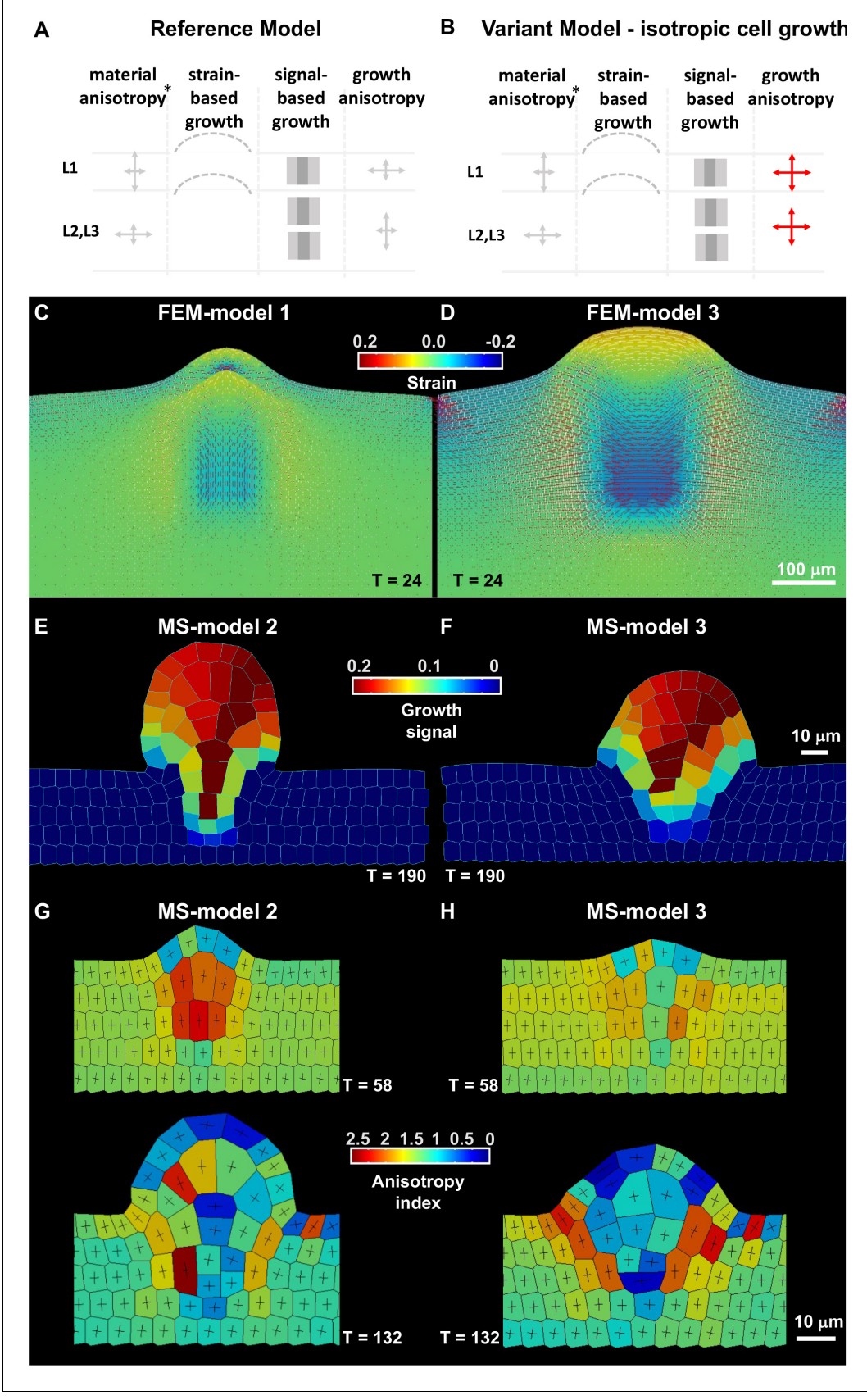

**Figure 3.** Mechanical and cell-based 2D simulation models of ovule primordium development predict that ovule

*Figure 3 continued on next page*

*Figure 3 continued*

shape depends on cell growth anisotropy. (**A–B**) Schematic representation of the main parameters as described in *Table 2* used for the reference model and variation models shown in C and *Figure 3—figure supplement 1*. The L1 and underlying L2,L3 layers are represented, arrows of different size indicate anisotropy, the bulged dotted lines represent the ability of the tissue to (passively) grow under strain, the gray fields represent the initial domain of the growth signal (pale gray) and the domain of fixed, high concentration (dark gray square), *, for FEM-simulations only. Deviations of these parameters are shown in red. (**C–D**) Tissue-based FEM simulations of ovule primordium growth. (**C**) Growth stability is reached at T = 24 in the reference FEM Model 1. (**D**) Simulations omitting the anisotropic growth parameter (FEM-Model 3) show abnormal ovule dome shape at the same simulation time. The magnitude of accumulated strain is indicated by the background color (according to the heatmap), while the principal strain directions are shown as fine lines (white: positive strain, corresponding to stretch, red: negative strain, corresponding to compression). (**E–H**) Cell-based MS simulations of ovule primordium growth showing growth signal distribution and anisotropy index at the indicated simulation times (**T**). (**E**) Reference Model (MS-Model 2) showing a realistic primordium shape with straight flanks, sharp curvature at the apex, and a narrow base at T = 190. (**G**) The reference model shows the emergence of a large, anisotropic cell with trapezoidal shape in the L2 at T = 58, confirmed at T = 132. (**F**) Simulation with isotropic cell growth in inner L2,L3 layers (MS-Model 3) produces a primordium with enlarged apex and basis and flatter dome at the same simulation time as in the Reference Model. (**H**) L2 cells in MS-Model 3 show reduced anisotropy as compared to the Reference Model. See also *Figure 3—figure supplement 1*, *Table 2*, Appendix 1 for modeling hypotheses and methods.

The online version of this article includes the following video and figure supplement(s) for figure 3:

**Figure supplement 1.** FEM- and MS-based simulations, reference models, and additional variations.

**Figure 3—video 1.** Mass Spring Model 2 (reference model) compared against Mass Spring Model 5 (epidermal driven growth).

https://elifesciences.org/articles/66031#fig3video1

**Figure 3—video 2.** Mass Spring Model 2 (reference model) compared against Mass Spring Model 3 (isotropic growth).

https://elifesciences.org/articles/66031#fig3video2

---

anisotropy (for FEM models only). In addition, in the MS-based model, growth was prescribed to occur exclusively along the periclinal direction in the L1, according to our observations.

Then, using the versatility of the modeling framework to vary initial conditions, we tested the influence of the spatial distribution of the specified growth signal on primordium growth. As first variation, we let the growth signal diffuse broadly in the domain while maintaining the selected initial cells at a prescribed high intensity of the growth signal (FEM-Model 4, MS-Model 4). The emerging primordium is appreciably broader than the Reference Model (*Figure 3—figure supplement 1B*). We then explored the contribution of the growth signal in the L1 compared to the inner L2,L3 tissue to primordium growth. In FEM-Model 5 and MS-Model 5, the prescribed high growth signal is present only in the L1 but is free to diffuse to the L2,L3 layers. This produced a sharp primordium, narrower and taller than the primordium in the Reference Model, and a thicker L1 layer in the FEM model (*Figure 3—figure supplement 1B*, *Figure 3—video 1*). The L2 apical domain is narrower as compared to the Reference Models. When the growth signal is absent in the inner L2,L3 tissue (FEM-Model 6, MS-Model 6), signal growth in the L1 alone is not sufficient to enable primordium growth (*Figure 3—figure supplement 1B*).

To answer the complementary question whether a growth signal is required in the L1 for primordium growth when it is present in the L2,L3 layers, it was removed in FEM-Model 7 and MS-Model 7 (*Table 2*; *Figure 3—figure supplement 1B*). This scenario indeed enables growth of a digit-shaped structure (as long as strain-based growth is permitted), yet the dome appears shallower than in the Reference Model in FEM simulations while it is comparable in MS simulations.

To conclude, both models where a high level of growth signal is selectively present in L1 or in inner L2,L3 layers can produce a digit-shaped primordium. Yet, absence of a growth signal in the inner L2,L3 layers results in drastic shape alterations. This favors a scenario where inner L2,L3 layer-driven growth plays a fundamental role.

Next, through modulation of passive strain-based growth, we determined that a broad tissue domain uniformly competent for strain accommodation is necessary to resolve the high accumulation of stress, which limits primordium elongation (FEM-Model 8, MS-Model 8, *Figure 3—figure supplement 1C*). We also explored the contribution of material anisotropy for FEM-based simulations

(FEM-Model 2 and FEM-Model 2a, *Figure 3—figure supplement 1C*): even in the case of isotropic material, it is possible to grow a digit-shaped protrusion, yet with a wider dome and increased L1 thickness, contrasting experimental observations. To restore L1 thickness, it is sufficient to prescribe material anisotropy exclusively in the L1 (FEM-Model 2a).

Finally, we asked whether growth anisotropy must be specified in the model or if organ geometry and mechanical constraints are sufficient to specify primordium shape. Removing the growth anisotropy component abolished the digit shape of the primordium and produced a hemi-spherical protrusion in both FEM- and MS-based models (*Figure 3B–F*, *Figure 3—video 2*).

In summary, we identified parsimonious growth principles shaping the ovule primordium and suggesting different contributions of the epidermis and inner layers: an active tissue growth, mostly inner L2,L3 layer-driven and requiring a narrow, pit-shaped domain of a growth-signal, complemented by passive tissue growth with a necessary response of the L1 to accommodate the accumulated strain. Furthermore, material anisotropy in the L1 is predicted to play a role in constraining L1 thickness as observed experimentally. The fact that two distinct modeling approaches converge on the same morphogenetic principles suggests that the proposed growth mechanisms are robust. Furthermore, when compared to the growth dynamics of real ovules from phase 0-I to phase 1-I, the cell-based MS model showed a good agreement (*Figure 3—figure supplement 1D*).

Next, we used cell-based MS simulations to explore the correlation between organ growth and SMC morphological differentiation. In the Reference Model (MS-Model 2), an L2 apical cell with a trapezoidal shape, elongated along the main direction of ovule growth, emerged consistently during simulation (note that these cells still divide in the models as no special rule has been assigned to them) (*Figure 3D*). These are similar to SMC candidates at stage 0-II in real primordia, in a 2D longitudinal, median section through the ovule (*Figure 2K*). The elongated-trapezoidal shape of such a cell is not a prescribed feature of the model, but rather emerges from the combination of assigned anisotropic cell growth and geometrical constraints imposed by the surrounding, growing tissues.

Next, we explored the role of cell growth anisotropy in ovule primordium shape and SMC emergence. The MS-Model 3 (*Figure 3E–F*) corresponds to a virtual mutant where cell growth is isotropic in inner layers (the L1 maintained anisotropic growth to preserve its thickness). This led to an ovule primordium with a wider and flatter dome, comparable to FEM-Model 3. Despite the absence of a specified growth direction in inner L2,L3 tissue, due to geometrical constraints, the primordium still grows mostly vertically. We wanted to assess whether such geometrical constraints enable the formation of an elongated, trapezoidal SMC candidate also in the case of prescribed isotropic growth. As *Figure 3G–H* shows, the SMC candidate does not display the stereotypical shape and is not even elongated. Yet, mild cell anisotropy can be reached if the SMC candidate is allowed to grow twice more than in the Reference Model (MS-Model 3a, *Figure 3—figure supplement 1E*).

Altogether, the different simulations suggest that the anisotropy and characteristic shape of the SMC could be an emerging property of ovule primordium growth connected to geometrical constraints, even in the absence of specified anisotropic growth. The complementary model, altering cell growth anisotropy specifically in the L1, further suggested that directional cell growth in the epidermis is necessary to accommodate inner L2,L3 layer-driven growth and permit primordium elongation (MS-Model 3b, *Figure 3—figure supplement 1F*).

In all simulations, the candidate SMC eventually divided as the models miss a causative rule to differentially regulate cell division. When we prevented the SMC candidate to divide, its enlargement overrode primordium shape control in our simulations, creating an enlarged dome (*Figure 3—figure supplement 1G*). This suggests that a mechanism may limit SMC growth in real ovule primordia.

Altogether, 2D mechanical simulations in both continuous tissue-based (FEM) or cell-based (MS) approaches confirmed a role for localized cell growth compatible with experimental observations. The simulations also pointed to the importance of a differential role for the epidermis (accommodation) and the inner L2,L3 tissue (major growth component) as well as anisotropic cell growth as necessary for ovule primordium shape. Furthermore, the simulations suggested that SMC formation can be an emerging property of primordium geometry.

### *Katanin* mutants show a distinct ovule primordium geometry

To experimentally test the prediction of the isotropic growth models, we analyzed ovule primordium growth and SMC fate establishment in *katanin* (*kat*) mutants with well-described and -understood geometric defects: in absence of the microtubule-severing protein KATANIN, the self-organization

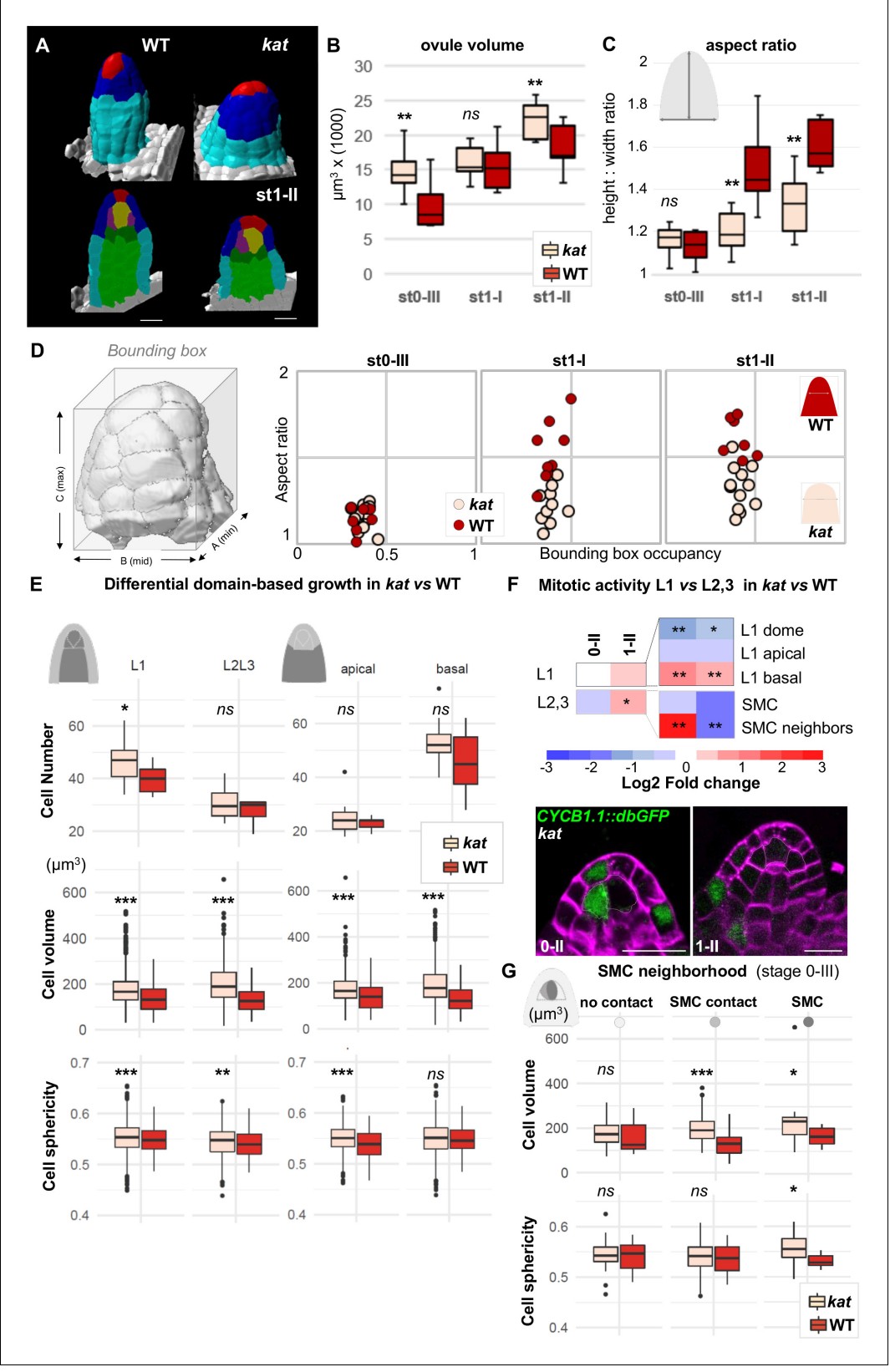

**Figure 4.** *katanin* mutants show a distinct ovule primordium geometry. Comparison between wild-type (WT, Ws-4 accession) and *katanin (kat, bot1-7* allele) ovule primordia. (**A**) 3D segmented images at stage 1-II. External organ view (top) and longitudinal sections (bottom) are shown. Scale bar 10 μm. See *Figure 4—source data 1* for full

*Figure 4 continued on next page*

*Figure 4 continued*

datasets. (B–D) Morphological difference between WT and *kat* primordia measured by their volume (B), the aspect ratio (C), and aspect ratio to bounding box occupancy relationship (D). (D), Left: scheme representing the bounding box capturing the primordium's 3D surface. See also *Figure 4—figure supplement 2A*. (E) Quantification of cell number, cell volume, and sphericity in comparisons of L1 *vs.* L2,L3 and apical *vs.* basal domains at stage 0-III. See also *Figure 4—figure supplement 2C*. (F) Mitotic activity domains are altered in *kat* primordia. Top: Heatmap of Log2 fold change of mitotic activity in the *kat* mutant (*lue1* allele) *vs.* WT (Col-0) *per* domain, at two developmental stages. The frequency of mitoses was measured as in *Figure 2*. Full maps of mitotic activity in different mutant alleles are shown in *Figure 4—figure supplement 2E*. Bottom: representative images of *kat* primordia (*lue1* allele) expressing CYCB1.1db-GFP. Dashed lines mark L2 apical cells. Magenta signal: Renaissance SR2200 cell wall label. Scale bar 10 µm. (G) Mean cell volume and sphericity are increased in *kat* L2 apical cells in contact with the SMC. SMC, cells in contact with the SMC, and cells not in direct contact with the SMC, are compared at stage 0-III. Color code in all plots: Dark red: WT; Salmon: *kat* mutant. Error bars: standard error of the mean. Differences between WT and *kat* mutants in (B), (C), (E), and (G) were assessed using a Mann Whitney U test; a two-tailed Fischer's exact test was used in (F). p values: *p≤0.05, **p≤0.01, ***p≤0.001. See also *Figure 4—figure supplements 1* and *2*; and *Figure 4—source data 2*.

The online version of this article includes the following source data and figure supplement(s) for figure 4:

**Source data 1.** Image gallery.
**Source data 2.** Raw data for quantitative analysis.
**Figure supplement 1.** KATANIN localization pattern in ovule primordium.
**Figure supplement 2.** *katanin* ovule growth defects.

of cortical microtubules in parallel arrays is hindered, thereby decreasing the cellulose-dependent mechanical anisotropy of the cell wall, resulting in more isotropic growth (*Bichet et al., 2001*; *Burk and Ye, 2002*). Note that *KAT* is expressed ubiquitously and, thus, also in the ovule: the KATANIN protein could be detected in both epidermal and internal L2,L3 tissue layers using a GFP reporter (*Figure 4—figure supplement 1*).

Here, we specifically analyzed the shape and cellular organization of ovule primordia in *kat* mutants using the *botero* (*bot1-7*) (*Ws* background), *lue1*, and *mad5* alleles (*Col* background) (*Bichet et al., 2001*; *Bouquin et al., 2003*; *Brodersen et al., 2008*). We generated and analyzed a new dataset of 59 annotated 3D digital *kat* and corresponding wild-type ovule primordia at stages 0-III, 1-I, and 1-II (*Figure 4—source data 1*). *kat* mutant primordia clearly showed an increased size and a more isotropic shape (*Figure 4A*), being 1.5 times bigger in volume than wild-type primordia (p=0.007, stage 0-III, *Figure 4B*) with a smaller aspect ratio (p<0.01 stages 1-I, 1-II, *Figure 4C*). Because the width-to-height ratio does not inform on the shape at the flanks, we derived an equation to estimate the 'plumpiness' of the primordia: primordia with rounder flanks will have a higher bounding box occupancy, that is, the volume fraction of a fitting, 3D parallelepiped (bounding box) effectively occupied by the primordium (*Figure 4D*, left), than straight digit-shaped ovules of the same aspect ratio. Mutant primordia are clearly distinct from wild-type primordia in their relationship between aspect ratio and bounding box occupancy at stage 1-I (*Figure 4D*, *Figure 4—figure supplement 2A*), when the primordium normally starts elongating along the major growth axis. These measurements confirmed a marked attenuation of anisotropic growth in *kat* ovule primordia as was observed in roots, shoot organs, or seeds (*Bichet et al., 2001*; *Hervieux et al., 2016*; *Luptovčiak et al., 2017b*; *Ren et al., 2017*; *Uyttewaal et al., 2012*; *Wightman et al., 2013*; *Zhang et al., 2013*). Yet, at a global level, the mean cell number, cell volume, and sphericity are not significantly different between *kat* and wild-type primordia (*Figure 4—figure supplement 2B–C*). Thus, these global approaches did not resolve the origin of primordia mis-shaping.

To refine the analysis, we contrasted different layers as done for wild-type primordia and found local alterations that provide an explanation for the formation of broader and more isotropic primordia in *kat* mutants. Indeed, at stage 0-III, *kat* ovules display a broader epidermis domain composed of more and larger cells than wild-type ovules (p=0.03 and p<0.001, respectively) (*Figure 4E*). Mitosis frequency analysis indicated a shift in cell division from the apex toward the basis (2.4 times less mitoses in the L1 dome domain and 2.7 times more in the L1 basal domain, compared to the wild type) (*Figure 4F*, *Figure 4—figure supplement 2E*), consistent with the increased cell number observed in L1. Yet increased cell division is not a general characteristic of *kat* primordia since, overall, the apical and basal domains are not massively overpopulated. By contrast, *kat* cells are generally

larger and slightly more spherical in all domains (*Figure 4E*, *Figure 4—figure supplement 2C–D*). When specifically looking at the L2 apical domain, where the SMC differentiates, we noticed an increased relative frequency of CYCB1.1db-GFP expression in *kat* SMC neighbors at stage 0-II, whereas it decreases at stage 1-II (*Figure 4F*, *Figure 4—figure supplement 2E*). This is in stark contrast with SMC neighbors in wild-type primordia displaying first a relative mitotic quiescence at stage 0-II, and then enhanced mitotic activity at stage 1-II. Yet, in the SMC, no mitotic activity increase was observed in *kat* as compared to wild-type primordia (*Figure 4F*, *Figure 4—figure supplement 2E*).

In conclusion, the absence of *KAT*-mediated cell growth anisotropy is associated with spatio-temporal shifts in cell divisions, leading to an altered primordium geometry including a flatter dome and enlarged basis. Interestingly, these changes coincided with the occurrence of additional large cells in the L2 apical domain of *kat* primordia (SMC direct neighbor cells, *Figure 4G*, Materials and methods) similar in size to the SMC. This raises the question of the identity of these ectopic, enlarged neighbor cells.

## Altered ovule primordium geometry in *katanin* mutants induces ectopic SMC fate

Following the observation of mis-shaped *kat* primordia that contained larger L2 apical cells, we asked whether this had an impact on cell identity and SMC establishment. As in the wild type, a clear SMC is identifiable in *kat* primordia, yet it appears slightly bigger and more spherical with some variability over stages (*Figure 5A*, *Figure 5—figure supplement 1*). Interestingly, and consistent with the increased mitotic frequency in SMC neighbors observed at earlier stages, at stage 1-II, we scored additional SMC neighbors in *kat* primordia. On average, these were larger (14%, p=0.006) and slightly more isotropic in shape (~11% less ellipsoid p=0.03, ~2% more spherical, p<0.01) than in the wild type (*Figure 5B*). At stage 1-II, 34% of the *kat* primordia (n = 109) showed more than one enlarged, subepidermal cell; decreasing to 14% at stage 2-II (n = 104), in stark contrast with wild-type primordia showing a majority (86% to 96%) of primordia with an unambiguous, single SMC (*Figure 5C*). The *kat* phenotype is thus reminiscent of mutants affecting SMC singleness (*Pinto et al., 2019*) and we hypothesized that these enlarged cells are ectopic SMC candidates.

To verify this hypothesis, we introgressed several markers of SMC identity in a *kat* mutant background. The first marker is a GFP-tagged linker histone variant (H1.1-GFP) that marks the somatic-to-reproductive fate transition by its eviction in the SMC at stage 1-I (*She et al., 2013*). H1.1-GFP eviction occurred in more than one cell in 29% of *kat* primordia (n = 43, *Figure 5D*). The second marker reports expression of the *KNUCKLES* transcription factor (KNU-YFP) in the SMC (*Tucker et al., 2012*). Detectable as early as stage 1-I in the wild type, it was ectopically expressed in 21% of *kat* primordia at stage 1-II (n = 43) (*Figure 5E*, *Figure 5—figure supplement 2A*). Third, to test whether the ectopic SMC candidates have a meiotic competence, we analyzed the AtDMC1-GUS reporter (*Agashe et al., 2002*; *Klimyuk and Jones, 1997*) and, indeed, scored ectopic expression in 57% of *kat* primordia (n = 56) (*Figure 5F*). However, using aniline blue staining of callose deposition, which in wild-type SMCs stains the cell wall immediately before meiosis and marks the cells walls of tetrads after meiosis, we never detected ectopic cells accumulating callose or ectopic tetrads in *kat* ovules (n = 119 and 269 ovules in *bot1-7* and *lue1*, respectively) (*Figure 5—figure supplement 2B*). We also noted a significant frequency of *kat* ovules lacking callose in SMCs or tetrads in both *kat* alleles, suggesting altered cell wall composition in the SMC, consistent with previous reports on *kat* somatic tissues (*Burk and Ye, 2002*) We next analyzed the *pWOX2-CenH3-GFP* marker labeling centromeres starting at the functional megaspore stage (FG1 stage) (*De Storme et al., 2016*) and found 21% *kat* primordia (n = 29) with two labeled cells, as compared to 4% (n = 26) in the wild type (*Figure 5—figure supplement 2C*). These ectopic spores are haploid since they display five centromeres. They may correspond to ectopic surviving spores given their alignment with the basal-most functional megaspore (*Figure 5—figure supplement 2C*). Using an additional gametophytic reporter, pAKV:H2B-YFP (*Schmidt et al., 2011*), we confirmed a significant number of ectopic spores at stage FG1 (31% in *kat vs.* 11% in wild-type ovules) showing a residual signal (*Figure 5—figure supplement 2D*). However, at later stages (FG2-FG7), we did not find evidence for ectopic embryo sacs (*Figure 5—figure supplement 2D*). At the FG7 stage, notable alterations in ovule morphology and the number of nuclei in the embryo sac were visible in *kat* as previously reported (*Luptovčiak et al., 2017b*).

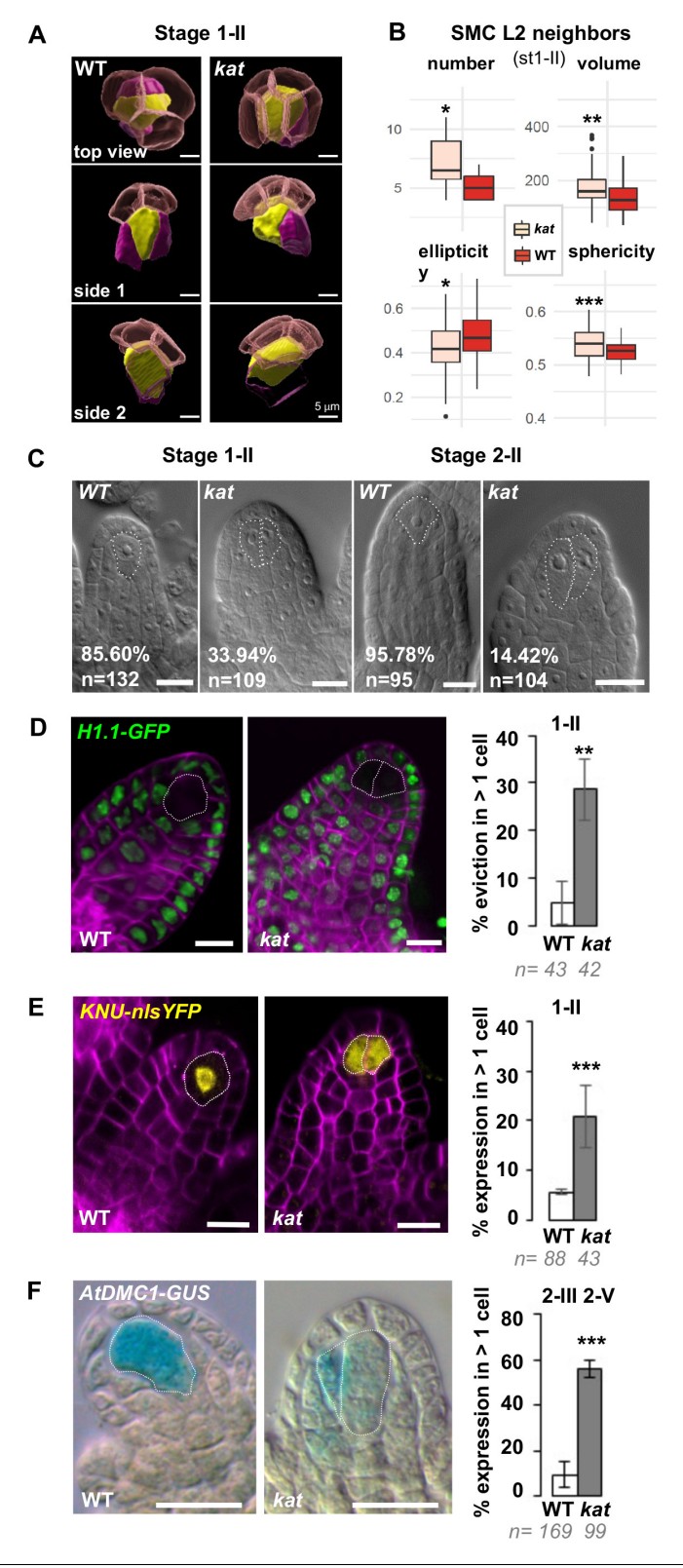

**Figure 5.** Altered ovule primordium geometry in *katanin* mutants is associated with multiple SMCs. (**A**) SMCs lose their typical pear shape in *kat* mutants. 3D images of the apical-most cells in top and side view as indicated, showing the SMC (yellow), SMC neighbors (purple), and the L1 dome (transparent red). (**B**) Differential properties of L2,L3 apical domain cells (SMC and SMC neighbors) in terms of cell number, mean cell volume, ellipticity, and

*Figure 5 continued on next page*

*Figure 5 continued*

sphericity at stage 1-II. See also *Figure 5—figure supplement 1* and *Figure 4—source data 2*. (C) Representative images of cleared wild-type (WT) and *kat* ovule primordia. The % indicate the frequency of ovules showing one SMC for WT primordia, or multiple SMC candidates (dashed lines) for *kat* primordia. (D–E) Representative images and quantification of SMC fate markers in WT and *kat* primordia: eviction of the *H1.1::H1.1-GFP* marker (green, D) and ectopic expression of *KNU::nls-YFP* (yellow, E) in more than one cell *per* primordia, are increased in *kat* primordia. See also *Figure 5—figure supplement 2A*. (F) The meiotic marker *AtDMC1::GUS* is ectopically expressed in *kat* ovules. Mutant alleles: *bot1-7* (A–B), *mad5* (C–E), *lue1* (F). Additional *kat* alleles, stages, markers, and detailed quantifications are presented *Figure 5—figure supplement 2* and *Figure 5—source data 1*. Magenta signal in (B) and (C): Renaissance SR2200 cell wall label. Scale bars for (A): 5 µm; for (C), (D), (E), (F): 10 µm. n: number of ovules scored. Error bar: standard error of the mean. Differences between WT and *kat* genotypes were assessed using a Mann Whitney U test in (B), and a two-tailed Fischer's exact test in (C), (D), (E), and (F). p values: *p≤0.05, **p≤0.01, ***p≤0.001. See also *Figure 5—figure supplements 1* and *2* and *Figure 5—source data 1*.

The online version of this article includes the following source data and figure supplement(s) for figure 5:

**Source data 1.** Raw data for quantitative analysis.
**Figure supplement 1.** Cell volume and sphericity of SMC and SMC neighbors in *katanin*.
**Figure supplement 2.** SMC and female gametophyte identity markers in *katanin*.

Taken together, these results show that ectopic, abnormally enlarged SMC neighbors in *kat* primordia show at least some characteristics of SMC identity, yet they do not complete meiosis and are likely reincorporated into the soma. After meiosis, ectopic surviving spores are observed but do not complete gametogenesis. Hence, reproductive fate establishment is altered in *kat* ovules but the defects do not persist beyond meiosis suggesting a regulative process.

## SMC singleness is progressively resolved during primordium growth

Based on our analysis of *kat* primordia, it appeared that the frequency of ectopic SMC candidates was high at early stages but decreased over time (*Figure 5C*). This is reminiscent of the phenotypic plasticity in SMC differentiation observed, to a lesser degree, in different *Arabidopsis* accessions (*Rodríguez-Leal et al., 2015*). To characterize this plasticity during development, we analyzed 1276 wild-type primordia from stage 0-II to 2-II and scored the number of primordia with one or two enlarged, centrally positioned, subepidermal cells (Classes A and B, respectively, *Figure 6A–B*). In the wild type, the majority of ovules showed a single candidate SMC (Class A) at stage 0-II but 27% of primordia (n = 289) had two SMC candidates (Class B), this frequency decreasing to 3% at stage 2-II (n = 103). This finding is consistent with ~ 5% primordia in wild-type at stage 1-II showing H1.1-GFP eviction (n = 43) and KNU-YFP expression (n = 88) in more than one cell, respectively (*Figure 5D–E*). Thus, instead of being immediate, SMC singleness can arise from a progressive restriction of fate among several SMC candidates during development. In wild-type primordia, SMC singleness is largely resolved at stage 1-I (*Figure 6B*).

Next, we quantified Class A and B primordia in *kat* mutants by scoring 2587 ovules of three different mutant alleles (*Figure 6C*, *Figure 6—figure supplement 1A*). Clearly, plasticity is higher with 42% (n = 202) class B primordia at stage 0-II in *kat* compared to 26% (n = 289) in the wild type. In addition, the resolution process is delayed in *kat* mutants as up to 33% Class B primordia persist at stage 1-II (n = 113) compared to 12% in the wild type (n = 281). Consistently, the two SMC markers used previously, clearly identify ectopic SMC candidates in *kat* primordia at stage 1-I (*Figure 6—figure supplement 1B,C*), and more significantly at stage 1-II (28.5% ectopic eviction of H1.1-GFP and 21% ectopic expression of KNU-YFP, *Figure 5D–E*). In addition, at stage 2-II, when 17% of cleared *kat* primordia (n = 110) showed ectopic SMCs, H1.1-GFP eviction and KNU-YFP expression was only found in 8% (n = 25) and 7% (n = 74) of the primordia, respectively (*Figure 6—figure supplement 1B,C*). Therefore, molecular events associated with SMC fate are partially uncoupled from cell growth during the resolution of SMC singleness in *kat*.

Another outcome of this study is the early emergence of SMC candidates, based on cell size mostly and confirming our former 3D cell-based analysis that showed increased cell volume of the L2,L3 apical cells already at stages 0-I/0-II (*Figure 2J*). To corroborate this finding using a different criterion, we measured the nuclear area, which is a distinctive feature of SMCs (*Rodríguez-*

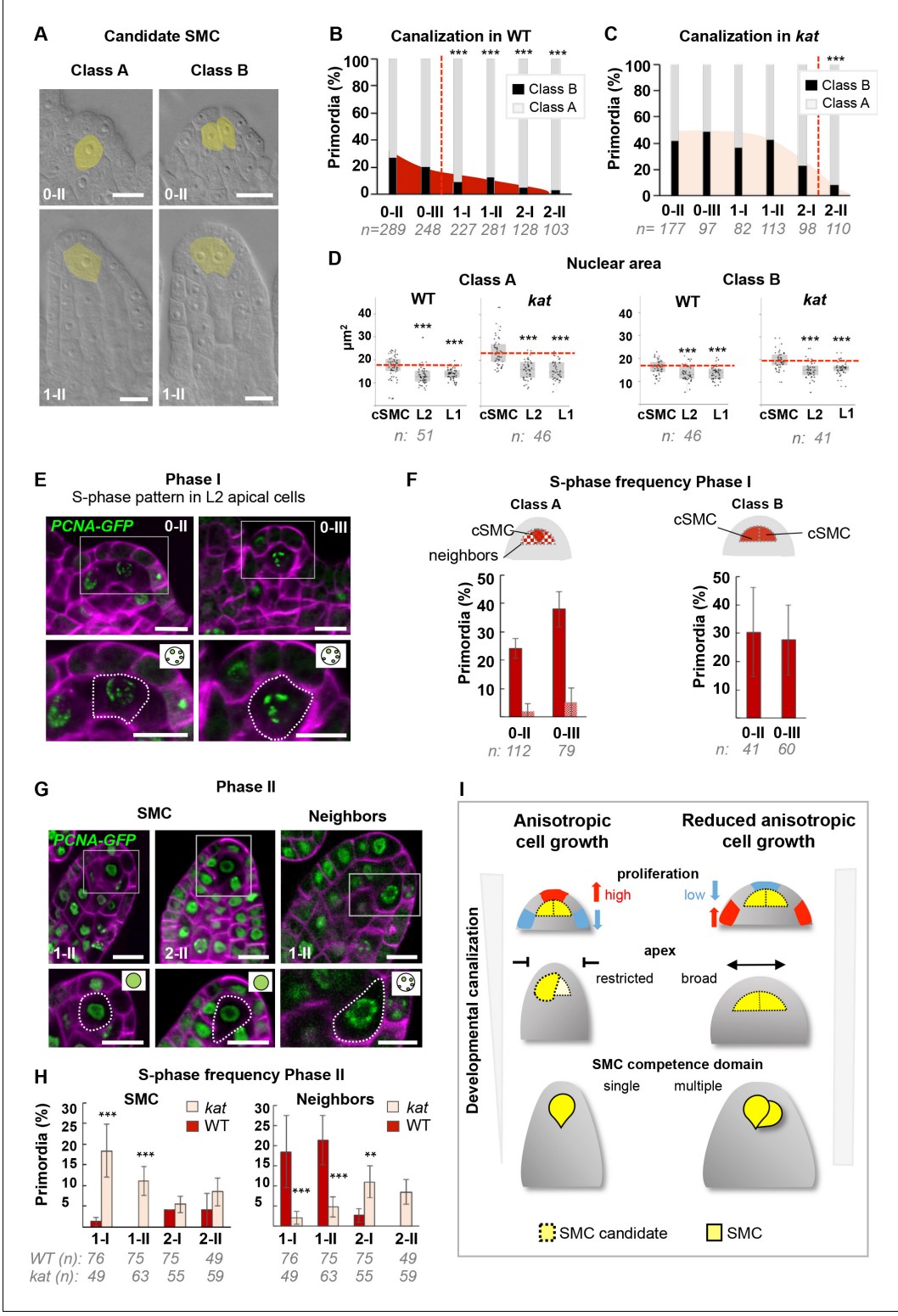

**Figure 6.** SMC singleness is progressively resolved during primordium growth. (**A–B**) In wild-type (WT) plants, ovule primordia harbor one (class A) or occasionally two (class B) SMC candidates with the frequency of class B gradually decreasing during development, reminiscent of developmental canalization. Typical images obtained by tissue clearing with SMC candidates highlighted in yellow (**A**) and plots showing the percentages of classes A and B (**B**). The frequency of class B ovules is significantly reduced from stage 1-I, suggesting that canalization (represented by the red dashed line) occurs before that stage. The plot coloration is a visual aid only.

*Figure 6 continued on next page*

*Figure 6 continued*

(**C**) Developmental canalization is delayed in *kat* mutants (*mad5* allele). The proportion of class B ovules is significantly reduced from stage 2-II only. Quantifications for two additional *kat* alleles are presented *Figure 6—figure supplement 1A*. The plot coloration is a visual aid only. (**D**) The candidate SMCs initially identified as enlarged cells consistently show an enlarged nucleus in both class A and class B primordia. Nuclear area is compared between the candidate SMCs (cSMC) and surrounding L2 and L1 cells (see *Figure 6—figure supplement 1D* and Materials and methods for further details). Box plots include jittered data points to visualize data variability. Red lines represent the median of cSMC nuclear area for comparison with other cell types. Equivalent quantifications for stage 0-III are presented in *Figure 6—figure supplement 1E*. (**E–F**) During Phase I (stages 0-II, 0-III), S-phase is detected in candidate SMCs at higher frequency than in neighbor cells in Class A ovules, and in both SMC candidates in Class B ovules. Representative images of ovule primordia showing the speckled S-phase pattern of the *pPCNA1::PCNA1:sGFP* reporter (green) (**E**). Magenta signal: Renaissance SR2200 cell wall label. Scale bar 10 μm. Quantification of speckled S-phase pattern in the SMC candidate and L2 neighbors (class A ovules) or in both SMC candidates (class B) (**F**). (**G–H**) During Phase II (stages 1-I to 2-II), in Class A wild-type ovule primordium, SMC exits S-phase, and neighbor cells undergo S-phase; while *kat* primordia show the opposite pattern. Representative images of Class A ovule primordia primarily showing the nucleoplasmic pattern of *pPCNA1::PCNA1:sGFP* in SMCs at stages 1-II (left panel) and 2-II (middle panel); and of the speckled S-phase pattern in neighbor cells (right panel). Quantification of speckled S-phase pattern in SMC candidates and neighbors in Phase II wild-type and *kat* primordia (**I**). Representative images and quantifications of Phase I *kat* primordia and of class B primodia are presented in *Figure 6—figure supplement 1F–H*. See also *Figure 2H*. (**I**) Model for the role of *KAT* in primordium growth and SMC differentiation (graphical abstract). In WT primordia, SMC differentiation follows a developmental canalization process influenced by cell growth anisotropy that shapes the primordium apex. In *kat* mutants, reduced anisotropy modifies the cell proliferation pattern, enlarges the apex and the L2,L3 apical domain, leading to multiple SMC candidates (delayed canalization). We propose that ovule primordium shape, controlled by anisotropic cell growth, determines SMC singleness. *Images*: scale bars: 10 um. *Graphs*: n = total number of ovules. Error bar: standard error of the mean (**F, I**). Differences between cell types, domains, or genotypes as indicated in the graphs were assessed using Wilcoxon signed rank test in (**D**), and a two-tailed Fischer's exact test in B, C, F, and I. p values: *p≤0.05, **p≤0.01, ***p≤0.001. Quantifications for additional alleles and detailed quantifications are provided in *Figure 6—figure supplement 1* and *Figure 6—source data 1*.

The online version of this article includes the following source data and figure supplement(s) for figure 6:

**Source data 1.** Raw data for quantitative analysis.
**Figure supplement 1.** SMC singleness is progressively resolved during primordium growth.

---

*Leal et al., 2015*; *She et al., 2013*). We compared the nuclear area of SMC candidates to that of surrounding L1 and L2 cells, in both Class A and B primordia, at stages 0-II and 0-III. Wild-type Class A and B primordia showed an enlarged nucleus in the candidate SMCs from stage 0-II onwards, and this correlation was also true in *kat* primordia (*Figure 6D*, *Figure 6—figure supplement 1D,E*).

However, it remains difficult to resolve the precise timing of SMC establishment at these early stages due to the lack of molecular markers. Yet, we reasoned that we may be able to distinguish the SMC candidates from their neighbors by their cell cycle pattern, where cells entering meiosis may engage in a specific S-phase compared to regularly cycling mitotic cells. To this aim, we used a GFP-tagged PCNA variant marking the replication machinery, *pPCNA1::PCNA1:sGFP* (PCNA-GFP) (*Yokoyama et al., 2016*). When engaged at active replication forks, PCNA-GFP shows nuclear speckles characteristic of S-phase (*Strzalka and Ziemienowicz, 2011*). During G1/G2, it remains in the nucleoplasm and in M-phase it is undetectable. We specifically quantified the distribution patterns of PCNA-GFP in cells from the L2 apical domain in wild-type and *kat* primordia, separately for Class A and B ovules (*Figure 6E–H*).

In Phase I wild-type primordia, PCNA-GFP was always detectable, in both classes, indicating that L2 apical cells are rarely in M-phase, consistent with the seldom detection of mitoses using CYCB1.1db-GFP. We observed an S-phase pattern consistently in one (Class A) or more (Class B), centrally positioned L2 apical cells, presumably corresponding to SMC candidates (*Figure 6E*). This pattern was captured in a large proportion of primordia at stage 0-II and 0-III: ~24% (n = 112) and ~40% (n = 79) for Class A, respectively; 30% (n = 41) and 28% (n = 60) for Class B, respectively (*Figure 6F*). Such high frequencies could be generated either by a slow S-phase in SMC candidates only (the persistence of the marker increasing the probability to score it repeatedly in our sample size), or by a regular (short) mitotic S-phase in a high number of SMC candidates. The low detection

frequency of the mitotic marker CYCB1.1db-GFP in SMC candidates is inconsistent with the latter possibility. Thus, the likeliest interpretation is that candidate SMCs enter a slow S-phase from early stages onwards, probably meiotic, although this cannot be assessed with this marker.

In *kat* primordia at Phase I, Class A ovules showed a wild-type pattern in the SMC at stage 0-II, but a reduction at stage 0-III, suggesting alterations in S-phase entry; and Class B ovules, by contrast, displayed high frequency of S-phase patterns, indicating either a longer S-phase or multiple dividing cells (*Figure 6—figure supplement 1G*).

In Phase II, SMC candidates showed no S-phase pattern in a large majority of both Class A and B primordia (97% on average, over entire Phase II) (*Figure 6G–H*, *Figure 6—figure supplement 1G–H*) in agreement with the presence of newly replicated DNA at stage 1-I (*She et al., 2013*). In contrast to Phase I, however, S-phase is now detected in SMC neighbors (~21% at stage 1-II) (*Figure 6G–H*), consistent with the divisions observed in these cells (*Figure 2D*). Strikingly, in Phase II, *kat* primordia display a higher frequency of S-phase patterns in SMC candidates (*Figure 6H*, *Figure 6—figure supplement 1F,H*), suggesting that S-phase duration or entry is delayed compared to the wild type. SMC neighbors, by contrast, show a lower frequency of S-phase patterns (Class A), consistent with reduced division in these cells (*Figure 4F*).

Collectively, our data indicate a cellular heterogeneity in terms of size, nuclear size, and S-phase patterns, of the L2 apical domain as compared to L1, which leads to the emergence of one or several SMC candidates as early as stage 0-II. The gradual decrease in the number of primordia with ambiguous SMC candidates demonstrates a developmentally regulated resolution of SMC fate to a single cell. This process is associated with a specific cell cycle progression, cellular elongation, and a robust expression of SMC fate markers. In *kat* primordia displaying alterations in geometry, SMC singleness is largely compromised: plasticity in SMC emergence is increased and fate resolution to a single SMC is delayed.

## Discussion

Organogenesis involves coordinated cell division and cell expansion, complex growth processes orchestrated by biochemical and mechanical cues (*Echevin et al., 2019*). How cell differentiation is coordinated in space and time during organ growth, and whether these processes are interrelated, are central aspects for the elucidation of patterning principles (*Whitewoods and Coen, 2017*). The female germline is initiated with SMC differentiation in the ovule primordium. The SMC emerges as a large, elongated, subepidermal cell that is centrally located at the apex of the primordium, a digit-shaped organ emerging from the placenta. To study how SMC fate relates to ovule organogenesis, we generated a reference collection of images capturing ovule primordium development at cellular resolution in 3D and determined cell division frequencies in the different domains. We observed a biphasic pattern of cell divisions alternating in the epidermis and inner layers, as well as the apical and basal domains, in Phase I (stages 0-I to 0-III) and Phase II (stages 1-I to 2-II).

However, this approach did not allow us to resolve the driving morphogenetic factors. For this reason, we developed continuous tissue-based and cell-based 2D simulations of primordium growth. The different simulations revealed key growth principles shaping the ovule primordium and uncovered differential roles for the epidermis and inner layers. Notably, an inner tissue-driven growth model, where the L1 also contributes to the expansion of the primordium, best described ovule primordium growth. This is reminiscent of a model describing leaf primordium emergence (*Peaucelle et al., 2011*). In addition, best-fit models produced by both cell- and tissue-based simulations predicted a growth-promoting signal in a confined domain along a vertical stripe at primordium emergence. Candidate growth signals are phytohormones, peptides, and small RNAs known to affect ovule primordium growth (*Kawamoto et al., 2020*; *Pinto et al., 2019*; *Su et al., 2020*). The domains of auxin response restricted in the L1 dome and of cytokinin signaling localized in a region basal to the SMC in Phase II primordia (*Bencivenga et al., 2012*) also suggest a confined growth signal. Whether the signaling domains are established already in Phase I and play a causative role in primordium patterning remains to be determined. The epidermis, by contrast, is predicted to play a key role in accommodating the constraints generated by inner-tissue growth. In this layer, strain-based growth and anisotropic material properties possibly resolve mechanical conflicts arising between tissue layers that grow at different rates (*Hervieux et al., 2017*). In line with this hypothesis,

we observed frequent divisions in L1 apical cells in vivo that support the expansion of the epidermis while inner tissues develop.

Interestingly, while our models did initially not contain an a priori rule to produce the typical, elongated shape of the SMC, it emerged consistently as a trapezoid-shaped L2 cell in the cell-based reference model. This shape likely emerges from the combination of assigned anisotropic cell growth and geometrical constraints imposed by the surrounding growing tissues. Explicitly blocking SMC division during the simulation, did not only enable its expansion as expected, but also pushed surrounding cells and strongly deformed ovule morphology. Thus, ovule growth homeostasis in vivo likely requires a mechanism to accommodate the differential growth of the SMC. A gradual reduction of SMC turgor pressure is a plausible scenario that would limit SMC size and prevent overriding the constraints provided by surrounding cells, similar to what was suggested for the shoot apical meristem (*Long et al., 2020*). In turn, a gradient of pressure in a field of cells could provide positional information through the directional movement of water and other molecules, thereby linking organ growth homeostasis, cell growth, and cell fate (*Beauzamy et al., 2014*; *Long et al., 2020*). Such a mechanism could also participate in determining the domain of the growth signal predicted by our simulations.

Another prediction of our growth models is the key role of anisotropic cell growth in controlling the geometry of the ovule primordium. Primordia of the *kat* mutant, deficient in the microtubule severing enzyme KATANIN (*Luptovčiak et al., 2017a*), resemble virtual mutant primordia generated by models where cell growth is isotropic in the inner layers. *kat* primordia have a flatter dome and large basis associated with global alterations in the cell proliferation pattern, cell size, and cell shape. Interestingly, *kat* primordia develop ectopic SMC candidates as early as stage 0-II. The most parsimonious hypothesis is that the altered geometry in *kat* primordia expands the domain of cells competent to form SMCs. Yet, we cannot exclude a direct effect of the *kat* mutation on L2 apical cells, disconnected from organ geometry, which would induce de novo SMC fate. However, this scenario is unlikely. First, KATANIN is present throughout ovule primordium cells. Second, we would expect increased cell growth and slower mitoses (*Luptovčiak et al., 2017a*), resulting in a reduced division frequency of L2 apical cells, which is not the case. Instead, we measured increased cell divisions in L2 SMC neighbors at stage 0-II. Also, the delayed divisions of SMC neighbors in *kat* at stage 1-II cannot explain the formation of ectopic SMC candidates at stage 0-II.

Therefore, the most parsimonious explanation is that the emergence of several SMC candidates in *kat* primordia is an effect of ovule primordium geometry. In this scenario, isotropic cell growth and altered mechanical constraints in the tissue acting from primordia emergence onwards, lead to divisions and patterning alterations that expand the domain of cells competent for SMC fate. This working model paves the way to explore the role of epidermal geometry in controlling regulators of the cell cycle and SMC fate in L2 apical cells. This is reminiscent of the epidermis in the shoot apical meristem, affecting the dome shape and acting on stem cell regulators in underlying layers (*Gruel et al., 2016*; *Savaldi-Goldstein et al., 2007*). The predicted role of the primordium epidermis to accommodate – and perhaps feedback on – underlying growth constraints is particularly interesting considering the role of mechanical cues on gene regulation (*Fal et al., 2017*; *Landrein et al., 2015*). The epidermis of the ovule primordium is also a known source of signaling cues (*Kawamoto et al., 2020*; *Pinto et al., 2019*; *Su et al., 2020*). In addition, phytohormones act themselves on KATANIN-mediated oriented cell growth and cell division (*Luptovčiak et al., 2017a*). In this context, it is possible that mis-shaping of *kat* primordia arises from a disrupted feedback altering the distribution pattern of the hypothesized growth signals.

Furthermore, our analyses unveiled new characteristics of the SMC establishment process. We found that SMC candidates emerge from within a mitotically quiescent L2 apical domain, consistent with the finding that the H3.1 histone variant HTR13 is evicted, marking cell cycle exit (*Hernandez-Lagana and Autran, 2020*). In addition, SMC candidates have a markedly long S-phase compared to surrounding cells. These observations are reminiscent of the animal germline where mitotic quiescence is a prerequisite for meiosis (*Kimble, 2011*; *Reik and Surani, 2015*). Also, early SMC candidates already display a typically large cell and nucleus size and an elongated shape aligned with the primordium growth axis. Collectively, we found that SMC characteristics are established much earlier than previously thought, that is, soon after primordium emergence (stages 0-II/0 III). Moreover, these characteristics frequently arise in more than one SMC candidate at Phase I, typically resolving into a single SMC at the onset of Phase II. This clearly documents a developmental sequence of plasticity

at SMC fate emergence and progressive resolution of SMC fate to a single cell. This is reminiscent of developmental canalization, which refers to the capacity of an organism to follow a given developmental trajectory in spite of disturbances (*Hallgrimsson et al., 2019*; *Scharloo, 1991*; *Waddington, 1942*). Cell fate canalization is well studied in animal systems where it is modulated by intercellular signal-based feedback mechanisms (*Heitzler and Simpson, 1991*), epigenetic regulation (*Pujadas and Feinberg, 2012*), and organ geometry (*Huang et al., 2020*; *Huang and Russell, 1992*; *Pujadas and Feinberg, 2012*; *Royer et al., 2020*). In plants, canalization is better known in the context of organogenesis during phyllotaxis and developmental robustness (*Godin et al., 2020*; *Lempe et al., 2013*). Our study expands the examples of canalization to the level of a cellular domain in the Arabidopsis ovule primordium. While L2 apical cells initially share the competence to form SMC candidates, leading to plasticity at SMC emergence, the progressive restriction of cell fate possibilities in the primordium apex ultimately leads to the specification of only one SMC committed to meiosis. Our results are in line with a formerly proposed canalization process operating during SMC establishment (*Grossniklaus and Schneitz, 1998*; *Rodríguez-Leal et al., 2015*). Despite the fact that several mutations (reviewed in *Pinto et al., 2019*; *Mendes et al., 2020*), including *kat* (this study), alter SMC singleness in Arabidopsis, canalization remains a robust process securing the formation of a single embryo sac despite such genetic perturbations. How this developmental mechanism buffers phenotypic inter-individual variations and whether it is evolutionary constrained remains to be determined.

In this study, we quantified plasticity among ovule primordia and progressive fate resolution during primordium growth in wild-type reference accessions. We characterized specific cellular events associated with these processes, notably differential cell growth and cell division. SMC fate emergence is characterized by mitotic quiescence and cellular growth in one or more L2 apical cells. SMC singleness resolution is associated with re-entry into a somatic cell cycle (this study), and re-incorporation of a replicative histone H3.1 (*Hernandez-Lagana and Autran, 2020*) in candidates neighboring the SMC. How known epigenetic and signaling factors interplay to secure SMC singleness remains to be determined. Similar to mouse and *Drosophila*, where tissue mechanics and organ geometry were shown to contribute to cell fate canalization (*Chan et al., 2017*; *Huang et al., 2020*; *Royer et al., 2020*), we propose that ovule primordium geometry contributes to channel SMC fate in the apex and the resolution into a single SMC. In this conceptual framework, *kat* increases plasticity and delays the resolution process toward SMC singleness (working model *Figure 6I*).

Altogether, our work proposes a conceptual framework linking organ geometry, cell shape, and cell fate acquisition in the ovule primordium, which is potentially of broader relevance in plant patterning. In addition, the image resource published in this study is complementary to others capturing ovule development at later stages (*Lora et al., 2017*; *Vijayan et al., 2021*). It also populates a growing number of 3D-segmented images of plant tissues and organs (*Wolny et al., 2020*), which collectively build the fundament of a developmental atlas integrating morphogenesis with gene expression (*Hartmann et al., 2020*).

## Materials and methods

### Key resources table

| Reagent type (species) or resource | Designation | Source or reference | Identifiers | Additional information |
|---|---|---|---|---|
| Chemical compound, drug | Renaissance SR2200 | Renaissance Chemicals | SCRI Renaissance Stain 2200 | https://www.renchem.co.uk/index.php/specialty-chemicals-division/item/48-selected-fluorescent-dyes-and-brighteners-for-microscopists |
| Chemical compound, drug | PI stain | Sigma- Aldrich | Catalog # P4170 | Propidium Iodide |
| Chemical compound, drug | Na-metabisulphite | Sigma- Aldrich | Catalog # S9000/PubChem: 329824616 | Sodium metabisulphite |
| Chemical compound, drug | Aniline Blue | Sigma- Aldrich | Catalog # 415049 | |

*Continued on next page*

Continued

| Reagent type (species) or resource | Designation | Source or reference | Identifiers | Additional information |
|---|---|---|---|---|
| Other | 3D digital atlas ovule primordium | This paper | https://doi.org/10.5061/dryad.02v6wwq2c | Data resource. 3D segmented images (.ims files) wild-type (*Col-0, Ws-4*) and *katanin* (*bot1-7*) as shown in *Figure 1—source data 1* and *Figure 4—source data 1* |
| Genetic reagent (*Arabidopsis thaliana*) | Col-0 | NASC | NASC (RRID:SCR_004576) Stock ID: N22625 | Wild-type ecotype Col-0 |
| Genetic reagent (*Arabidopsis thaliana*) | Ws-4 | NASC | NASC (RRID:SCR_004576) stock ID: N5390 | Wild-type ecotype Ws-4 |
| Genetic reagent (*Arabidopsis thaliana*) | *bot1-7* | doi:10.1046/j.1365-313x.2001.00946.x | | *bot1-7* mutant. H.Höfte. |
| Genetic reagent (*Arabidopsis thaliana*) | *mad5* | doi:10.1126/science.1159151 | | *mad5* mutant. O.Voinnet. |
| Genetic reagent (*Arabidopsis thaliana*) | *lue1* | NASC | NASC (RRID:SCR_004576) Stock ID: N57954 | *lue1* mutant |
| Genetic reagent (*Arabidopsis thaliana*) | *CYCB1db-GFP* | doi:10.1016/j.cub.2009.06.023 | | *promCYCB1.1::CYCB1.1db-GFP*. M.Bennett. |
| Genetic reagent (*Arabidopsis thaliana*) | *H1.1-GFP* | doi:10.1242/dev.095034 | | *promH1.1::H1.1-GFP*. |
| Genetic reagent (*Arabidopsis thaliana*) | *KNU-nlsYFP* | doi:10.1242/dev.075390 | | *promKNU::nls-YFP*. M.Tucker. |
| Genetic reagent (*Arabidopsis thaliana*) | *DMC1-GUS* | doi:10.1046/j.1365-313x.1997.11010001.x | | *promAtDMC1::GUS*. I.Siddiqi. |
| Genetic reagent (*Arabidopsis thaliana*) | pWOX2:CENH3-GFP | doi:10.1186/s12870-015-0700-5 | | *promWOX2::CenH3-GFP*. N.DeStorme. |
| Genetic reagent (*Arabidopsis thaliana*) | PCNA-GFP | doi:10.1038/srep29657 | | *promPCNA1::PCNA1-GFP*. S. Matsunaga. |
| Genetic reagent (*Arabidopsis thaliana*) | pAKV:H2B-GFP | doi:10.1371/journal.pbio.1001155 | | *promAKV::H2B-GFP*. W.C. Yang. |
| Genetic reagent (*Arabidopsis thaliana*) | GFP:KTN1 | doi:10.1126/science.1245533 | | *promKTN1::GFP-KTN1*. D.W. Ehrhardt. |
| Software, algorithm | IMARIS | http://www.bitplane.com/imaris/imaris | RRID:SCR_007370 | 3D image processing software Bitplane AG, Switzerland |
| Software, algorithm | ExportImarisCells, | This paper. | | plugin for IMARIS to export segmented cells in meshes for MorphographX. https://github.com/barouxlab/ExportImarisCells (copy archived at swh:1:rev:50badce519f07cf529c3abef765b58972fff70e6); *Mosca, 2021a*. |
| Software, algorithm | MorphoGraphX | https://morphographx.org/ | | Software to perform 2D/3D segmentation and image analysis |
| Software, algorithm | MorphoMechanX | https://morphographx.org/morphomechanx/ | | Software to perform 2D/3D biomechanical simulations |
| Software, algorithm | MassSpring Models_ovuleGrowth2D | This paper. | DOI:10.5281/zenodo.4681169 | plugin for MorphoMechanX, 2D MS-models simulation tool. https://github.com/GabriellaMosca/MassSpring_2DovuleGrowthModel (copy archived at swh:1:rev:a66b0496ba51ca7674f0020ace723aa0b850470f); *Mosca, 2021b*. |

*Continued*

| Reagent type (species) or resource | Designation | Source or reference | Identifiers | Additional information |
|---|---|---|---|---|
| Software, algorithm | FEM_2DOvule GrowthModel | This paper. | DOI:10.5281/zenodo.4681167 | plugin for MorphoMechanX, 2D FEM-models simulation tool. https://github.com/GabriellaMosca/FEM_2DOvuleGrowthModel (copy archived at swh:1:rev:6b6c09f6039ce3e9d55686d6b6e9dadf28b53e2f); *Mosca, 2021c*. |
| Software, algorithm | 3DAutoLabeling-ShapeQuant | This paper. | DOI:10.5281/zenodo.4681165 | plugin for MorphoMechanX, automatic labeling of L1, L2, L3 layers and cell shape quantifier. https://github.com/GabriellaMosca/3DAutoLabeling-ShapeQuant (copy archived at swh:1:rev:3138c3fb10891afbc1d48d35395dc00ee58ce199); *Mosca, 2021d*. |
| Software, algorithm | OvuleViz | This paper. | | R-based, shiny interface for interactive plotting of Imaris data. https://github.com/barouxlab/OvuleViz (copy archived at swh:1:rev:fd614aa1e80258928ee036191f26c3dd703d3141); *Pires, 2021*. |
| Software, algorithm | R Project for Statistical Computing/RStudio | https://www.r-project.org/ http://www.rstudio.com/ | RRID:SCR_001905 | *R Core Team, 2013* |
| Software, algorithm | OMERO | http://www.openmicroscopy.org/site/products/omero | RRID:SCR_002629 | *Besson et al., 2019* |
| Software, algorithm | FIJI | http://fiji.sc | RRID:SCR_002285 | *Rueden et al., 2017* |

## Plant growth and plant material

*Arabidopsis thaliana* plants were grown under long-day conditions (16 hr light) at 20–23°C in a plant growth room. Columbia (Col-0) and Wassileskija (Ws-4) accessions were used as wild-type controls depending on the mutant background used in the experiment. Three *katanin* alleles were used: *bot1-7* (*Bichet et al., 2001*) in Ws-4 accession, *lue1* (*Bouquin et al., 2003*) and *mad5* (*Brodersen et al., 2008*) both in the Columbia (Col-0) accession. Homozygous mutant individuals for all the *katanin* alleles were identified on the basis of their recessive vegetative phenotype. The following published markers were used: pCYCB1.1:db-GFP (*Ubeda-Tomás et al., 2009*), pKNU:nls:YFP (*Tucker et al., 2012*), pH1.1:H1.1:GFP (*She et al., 2013*), AtPCNA1:sGFP (*Yokoyama et al., 2016*), pAtDMC1:GUS (*Agashe et al., 2002*; *Klimyuk and Jones, 1997*), pWOX2:CenH3:GFP (*De Storme et al., 2016*), pAKV:H2B:GFP (*Schmidt et al., 2011*), and crossed to *kat* mutants, and to Ws-4 ecotype for *bot1-7* allele comparisons. For KATANIN localization, the published reporter line GFP:KTN1 in a *ktn1-2* background (*Lindeboom et al., 2013*) was used.

## 3D imaging and image processing (segmentation and labeling)

Entire carpels were stained using the pseudo-Schiff propidium iodide (PS-PI) cell wall staining procedure providing excellent optical transparency for 3D imaging in depth in whole-mount. We described previously the manipulation, staining, mounting of the flower carpels and imaging procedures (*Mendocilla Sato and Baroux, 2017*). Cell-boundary based image segmentation was done using ImarisCell (Bitplane) as described in details previously (*Mendocilla Sato and Baroux, 2017*). Each ovule was manually labeled in Imaris using customized Cell Labels for the different cell types and domains colored as shown in *Figure 1*. We defined the labels as follows:

- SMC (Spore Mother Cell): most apical central enlarged L2 cell. At stage 0-I, as enlargement is not always detected visually, the most apical L2 cell was then selected as candidate SMC (cSMC).
- L1: epidermal cells

- L1 dome: most apical cells in contact with SMC
- L2,L3: cells below the epidermis. L2 and L3 were not distinguished originally.
- Apical domain: group of cells at the apex of the primordium and encompassing the SMC and direct neighbor cells.
- Basal domain: group of cells at the basis of the primordium below the apical domain and until, but not including cells of the placental surface. At stages 0-I and 0-II only an apical domain is defined. A basal domain appears only starting stage 0-III.
- CC (Companion Cell): L2 cells in apical domain in contact with the SMC with an elongated shape (as judged in ovule longitudinal median section using the 'clipping plane' IMARIS tool).
- SMC contact: cells in contact with the SMC.

Semi-automated segmentation requiring user input and manual labeling can be error prone. To reduce the error rate, the 92 images were segmented and labeled by one author, but verified and curated by two others. All Imaris files used in the study are available at the DRYAD repository: https://doi.org/10.5061/dryad.02v6wwq2c.

## Ovule stage classification

Ovule development is described according to a well-accepted nomenclature (*Schneitz et al., 1995*). The first stage, initially defined as stage 1-I, indistinctly grouped primordia from emergence until digit shape. To enable describing early morphogenetic processes, however, we propose to (i) restrict stage 1-I to the final digit shape stage and (ii) subdivide preceding stages as stages 0-I, 0-II, and 0-III, as shown in *Figure 1B*. These developmental stages are classified according to the approximate number of cell layers protruding above the placenta and overall shape of the ovule as described in *Table 1* and *Figure 1—source data 1*.

## Quantification of cell number, size and shape and interactive plotting using OvuleViz

Several cell descriptors were retrieved using the ImarisCell' Statistics function and exported as. csv files: cell area, cell volume, cell sphericity, cell ellipticity (oblate and prolate). We developed an interactive R-based data plotting interface, OvuleViz, reading the Imaris-derived data within the exported files ordered by genotype then stages (*Figure 1—figure supplement 1*). OvuleViz is freely available at https://github.com/barouxlab/OvuleViz (*Pires, 2021*), and is based on a *shiny* interface for R. OvuleViz allows plotting selectively one or several of the cell descriptors for chosen stages and genotypes, along different visualization (scatter plots, box plots, histograms). In addition, OvuleViz retrieves the cell number from the number of objects in the. csv file.

## 3D quantification of ovule volume and shape using IMARIS

Ovule volume and shape were quantified on 3D segmentations using IMARIS software, to compare *katanin* and wild-type genotypes. For each ovule, all segmented labeled cells were duplicated and fused as a single cell object. The *ImarisCell' Statistics* function was used to retrieve 'cell volume' and 'bounding box OO' (object-oriented 3D bounding rectangle, exporting the minimum (A), mid (B) and maximum (C) lengths of the object bounding rectangle). Width to Height ratio was calculated by dividing the maximum (C) length by the mean of mid (B) + minimum (A) lengths. Bounding box occupancy was calculated as the ratio of ovule volume by the bounding box volume.

## Modeling and image analysis with MorphoMechanX

The details of tissue growth models and MorphoMechanX-based processing are described in Appendix 1.

## 2D cytological analysis of cleared ovule primordia and quantifications

For cytological examination of cleared ovule primordia, flower buds from wild-type and mutant plants were harvested and fixed in formalin-acetic acid-alcohol solution (40% formaldehyde, glacial acetic acid, 50% ethanol; in a 5:5:90 vol ratio) for at least 24 hr at room temperature. After fixation, samples were washed two times with 100% ethanol and stored in 70% ethanol. Gynoecia of 0.2–0.6 mm in length were removed from the flowers with fine needles (1 mm insulin syringes), cleared in Herr's solution (phenol: chloral hydrate: 85% lactic acid: xylene: clove oil in 1:1:1:0,5:1 proportions), and observed by differential interference contrast microscopy using a Zeiss Axioimager Z2

microscope and 40X or 63X oil immersion lenses. Picture acquisition was done with a sCMOS camera (Hamamatsu ORCA Flash V2). Nuclei area measurements were carried out with ImageJ software, using the manual contour tool 'Oval'.

## Fluorescence microscopy and quantifications

Epifluorescence imaging of callose using aniline blue staining was performed as described (*Cao et al., 2018*), and observed on a Zeiss Axioimager Z2 microscope with CFP emission filter, DIC, and a 63X oil immersion lens. Image acquisition was done with a sCMOS camera (Hamamatsu ORCA Flash V2).

Imaging of ovule primordia stained in whole-mount for cell boundary was done as described (*Mendocilla Sato and Baroux, 2017*) using a laser scanning confocal microscope Leica LCS SP8 equipped with a 63X glycerol immersion objective and HyD detectors.

Imaging of the GFP and YFP markers was performed using a laser scanning confocal microscope Leica LCS SP8 equipped with a 63X oil immersion objective, or a Leica SP5 equipped with a 63X glycerol immersion objective and HyD detectors. Samples were mounted in 5% glycerol with the cell wall dye Renaissance 2200 (SR2200) diluted 1/2000 or as described (*Musielak et al., 2015*). The following wavelengths were used for fluorescence excitation (exc) and detection of emission (em): Renaissance: exc = 405 nm, em = 415–476 nm; GFP: exc = 488 m, em = 493–550 nm; YFP: exc = 514 nm, em = 590–620 nm. For KATANIN localization, GFP:KTN1 *ktn1-2* carpels were mounted in gelrite 0.2% supplemented with Renaissance (SR2200) diluted 1/1000 in water, and imaged immediately using a Zeiss LSM880 laser scanning microscope equipped with a 40X long distance water immersion objective and Airyscan detector, in the SR (super-resolution) acquisition mode. Fluorescence was collected using the following filters settings: 405 nm and 488/561/633 nm primary beam splitters for all channels, then BP420−480 + BP495-550 secondary filter for 488 nm excitation channel (GFP) and BP420−480 + LP605 for 405 nm excitation channel (Renaissance). Images were processed using the built-in Airyscan processing tool.

All images were processed using ImageJ (*Rueden et al., 2017*), OMERO (*Besson et al., 2019*), or Imaris (Bitplane, Switzerland) for contrast/intensity adjustments and maximum intensity projections where relevant.

The mitotic activity in both wild-type and *katanin* ovules was quantified by scoring the cells expressing *promCYCB1::dbCYCB1-GFP* (M phase reporter) on 3D stacks, in each ovule domain at each developmental stage. Only ovules showing at least one cell expressing *promCYCB1::dbCYCB1-GFP* were included in the analysis. At a given stage, the frequency of mitoses per domain is the ratio between the number of GFP-positive cells within a domain in all observed ovules, to the total number of GFP-positive cells found in all domains in that population of ovules.

To quantify ectopic expression of H1.1-GFP, KNU-nlsYFP, pWOX2:CenH3-GFP and pAKV:H2B-GFP markers, the percentage of ovules presenting expression (or eviction in the case of H1.1-GFP) of the marker in more than one cell (or in more than one embryo sac for pAKV:H2B-GFP) was scored on 3D stacks. To quantify PCNA-GFP patterns, using 3D stacks, cells of the L2 apical domain were classified according to the 'speckled' or 'nucleoplasmic' patterns, or absence of the marker.

## Histochemical detection of *uidA* reporter gene product (GUS staining)

Gynoecia of 0.4–0.6 mm in length were removed from the flowers with fine needles and placed in staining solution, using high stringency conditions for ferro- and ferricyanide concentrations to limit GUS product diffusion (0.1% Triton X-100, 10 mM EDTA, 5 mM ferrocyanide, 5 mM ferricyanide and 20 mg/ml 5 -bromo-4-chloro-3-indolyl-beta-d-glucuronic acid cyclohexyl-ammonium salt (X-gluc, Biosynth AG, Staad, CH) in 50 mM phosphate buffer), for 96 hr at 37°C. After staining, the samples were mounted in clearing solution (50% glycerol, 20% lactic acid diluted in water) and observed by differential interference contrast microscopy using a Zeiss Axioimager Z2 microscope. Picture acquisition was done with a color sCMOS camera (Axiocam 506 color Zeiss).

## Statistical analysis

To identify the main cellular descriptors – cell area, cell volume, sphericity, ellipticity oblate, ellipticity prolate – explaining variance of each cell types, at each ovule primordium growth stages and between genotypes, we used principal component analysis (PCA) on a cells' descriptors matrix. PCA

was executed using the R software version 3.6.3. In all different subsets of data, PCA was performed by singular value composition of the centred and scaled data matrix. Data entries with missing values were removed before analysis. All Cell data as exported by Imaris were uploaded but only cells with a 'Cell Label' as described in the method section '3D imaging' were plotted. For visualization of PCAs, only the first two principal components were represented in both score and loading plots.

To determine if the means between two datasets were significantly different we used two-tailed t-test when the data were normally distributed (n > 30). For all datasets with n < 30, we assume that normality was not possible to assess properly (as small samples most often pass normality tests), thus we used non-parametric tests. Wilcoxon Signed-Rank two-tailed test was used for paired quantifications, and Mann Whitney U two-tailed test was used for unpaired quantifications. Tests were performed in Excel or in R (wilcox.test function). To compare ovule proportions, we used two-tailed Fischer's Exact test, using https://www.langsrud.com/fisher.html, available online. Variability was assessed using the Standard Error of the mean (SE), indicated in the graphs and/or the supplemental data when applicable. For some datasets, boxplots were used to improve visualization of data distribution. The number of samples and biological replicates for each experiment are indicated in the figure and/or figure legends and/or supplemental data.

## Acknowledgements

We thank S Strauss (Max Planck Institute for Plant Breeding Research, Cologne, Germany) and B Lane (John Innes Centre, Norwich, UK) for providing code and assessment for mesh generation and simulations; S Guyer (Bitplane, Switzerland) and M Lartaud (CIRAD, France) for their help with IMARIS analyses; O Leblanc (IRD Montpellier, France) for his help with data visualization in R; the Montpellier imaging facility MRI, member of the national infrastructure France-BioImaging supported by the French National Research Agency (ANR-10-INBS-04) and the Center for Microscopy and Image Analysis of the University of Zurich (ZMB) for microscopy imaging infrastructures; M Ueda (Nagoya University, Japan), I Siddiqi (Center for Cellular and Molecular Biology, India), N de Storme (Katholieke Universiteit Leuven, Belgium), S Matsunaga (Tokyo Science University, Japan), M Tucker (University of Adelaide, Australia), R Dixit (Washington University in St. Louis, USA), DW Ehrhardt (Carnegie Institution for Science, Stanford, USA), WC Yang (Chinese Academy of Sciences University) and the NASC stock center for seeds. This work was funded by the University of Zürich, the IRD, a PRCI grant from the Agence Nationale de la Recherche/Swiss National Science Foundation (#ANR-16-CE93-0002/SNF-310030L_170167) to DA and CB; grants from the Swiss National Science Foundation (# 310030B_160336 to UG, # IZCOZ0_182949 to CB), from the Commission for Technology and Innovation (CTI grant #16997) to CB, from the Baugarten Stiftung Zürich to CB, a CONACYT fellowship (# 438277) to EHL, and a fellowship from the Forschungskredit of the University of Zurich (#FK-74502-04-01) to GM.

## Additional information

### Competing interests
Anja Frey: Anja Frey is affiliated with Novartis Pharma Schweiz AG. The author has no financial interests to declare. The other authors declare that no competing interests exist.

### Funding

| Funder | Grant reference number | Author |
| --- | --- | --- |
| Schweizerischer Nationalfonds zur Förderung der Wissenschaftlichen Forschung | 310030L_170167 | Célia Baroux |
| Agence Nationale de la Recherche | 16-CE93-0002 | Daphné Autran |
| Schweizerischer Nationalfonds zur Förderung der Wissenschaftlichen Forschung | 310030B_160336 | Ueli Grossniklaus |

| Schweizerischer Nationalfonds zur Förderung der Wissenschaftlichen Forschung | IZCOZ0_182949 | Célia Baroux |
| Kommission für Technologie und Innovation | 16997 | Célia Baroux |
| Baugarten Stiftung | | Célia Baroux |
| Consejo Nacional de Ciencia y Tecnología | 438277 | Elvira Hernandez-Lagana |
| Forschungskredit Universitaet Zuerich | FK-74502-04-01 | Gabriella Mosca |

The funders had no role in study design, data collection and interpretation, or the decision to submit the work for publication.

### Author contributions
Elvira Hernandez-Lagana, Conceptualization, Data curation, Formal analysis, Validation, Investigation, Visualization, Methodology, Writing - original draft, Writing - review and editing; Gabriella Mosca, Conceptualization, Data curation, Software, Formal analysis, Funding acquisition, Investigation, Methodology, Writing - original draft, Writing - review and editing; Ethel Mendocilla-Sato, Data curation, Formal analysis, Supervision, Methodology, Writing - review and editing; Nuno Pires, Software, Visualization; Anja Frey, Resources, Investigation; Alejandro Giraldo-Fonseca, Formal analysis, Visualization, Writing - review and editing; Caroline Michaud, Investigation; Ueli Grossniklaus, Supervision, Funding acquisition, Writing - review and editing; Olivier Hamant, Conceptualization, Resources, Supervision, Funding acquisition, Methodology, Writing - review and editing; Christophe Godin, Conceptualization, Funding acquisition, Writing - review and editing; Arezki Boudaoud, Conceptualization, Formal analysis, Supervision, Funding acquisition, Methodology, Writing - review and editing; Daniel Grimanelli, Conceptualization, Supervision, Funding acquisition, Methodology, Project administration, Writing - review and editing; Daphné Autran, Célia Baroux, Conceptualization, Resources, Data curation, Formal analysis, Supervision, Funding acquisition, Validation, Investigation, Visualization, Methodology, Writing - original draft, Project administration, Writing - review and editing

### Author ORCIDs
Elvira Hernandez-Lagana (iD) https://orcid.org/0000-0002-2645-3783
Gabriella Mosca (iD) http://orcid.org/0000-0003-4509-498X
Ethel Mendocilla-Sato (iD) http://orcid.org/0000-0003-0339-3535
Nuno Pires (iD) http://orcid.org/0000-0002-7113-3519
Caroline Michaud (iD) http://orcid.org/0000-0002-0620-2442
Ueli Grossniklaus (iD) http://orcid.org/0000-0002-0522-8974
Olivier Hamant (iD) http://orcid.org/0000-0001-6906-6620
Christophe Godin (iD) http://orcid.org/0000-0002-1202-8460
Arezki Boudaoud (iD) http://orcid.org/0000-0002-2780-4717
Daniel Grimanelli (iD) http://orcid.org/0000-0001-5424-114X
Daphné Autran (iD) https://orcid.org/0000-0002-5227-8966
Célia Baroux (iD) https://orcid.org/0000-0001-6307-2229

### Decision letter and Author response
Decision letter https://doi.org/10.7554/eLife.66031.sa1
Author response https://doi.org/10.7554/eLife.66031.sa2

## Additional files
### Supplementary files
• Transparent reporting form

## Data availability

The data generated or analysed during this study are included in the manuscript and supporting files. Source data files have been provided for all Figures. Segmented Image data are deposited in Dryad Digital Repository, available at: https://doi.org/10.5061/dryad.02v6wwq2c. Cell Statistics from these images are deposited as 'Segmented Dataset.csv' at https://github.com/barouxlab/OvuleViz (copy archived at https://archive.softwareheritage.org/swh:1:rev:fd614aa1e80258928ee036191f26c3dd703d3141).

The following datasets were generated:

| Author(s) | Year | Dataset title | Dataset URL | Database and Identifier |
|---|---|---|---|---|
| Sato EM, Autran D, Baroux C | 2021 | Arabidopsis ovule primordium stages 0-I to 2-II, wild-type Col_0, Ws-4 and botero (bot1-7) mutant | http://dx.doi.org/10.5061/dryad.02v6wwq2c | Dryad Digital Repository, 10.5061/dryad.02v6wwq2c |
| Sato EM, Autran D, Pires N, Baroux C | 2021 | Segmented_Dataset_Hernandez_Mosca_etal_2020 | https://github.com/barouxlab/OvuleViz/blob/master/Segmented_Dataset_Hernandez_Mosca_etal_2020 | GitHub, fd614aa |

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

# Appendix 1

## 1.1 Mass spring-based ovule growth simulations at cellular resolution

### 1.1.1 Template preparation and initialization - Regular grid of cells

Following observation of microscopy images of longitudinal sections of carpels showing the placenta, a regular grid of cells has been created with MorphoGraphX (*Barbier de Reuille et al., 2015*, https://morphographx.org/, Cell Maker plugin). The grid is composed of 28 columns by 5 rows of square cells of side 5.44 µm, the cell rows are staggered with respect to each other by 1.36 µm in the longitudinal direction.

The mass spring mesh is constituted by interconnected segments outlining the cells boundaries and representing the cell walls. The segments have a varying length of 1.36 µm and 2.72 µm, the top margin of the grid of cells has a segment length of 5.44 µm (this is to increase mass spring stability w.r.t. compressive forces generated during ovule growth). A dual mesh connecting the cell centers is associated to the mass-spring mesh.

The template initialization as well as the proper growth simulations are run with an in-house built plugin for MorphoMechanX (https://morphographx.org/morphomechanx/) available at a git repository: github.com/GabriellaMosca/MassSpring_2DovuleGrowthModel (*Mosca, 2021b*).

Once the template has been generated, it needs to be initialized with the parameters illustrated in *Figure 3—figure supplement 1B*. These properties are assigned at the cell level (no gradient for these properties within a cell), so on the dual graph.

The growth signal is the result of a discrete, cell-based diffusive process. The user sets some cells with fixed high concentration and some other cells with fixed lower concentration of the diffusive substance, while the other cells get their concentration as a result of the diffusive process. More specifically, for the simulations of the growth signal in this work, some cells were assigned a fixed concentration of 0.2 (the vertical column made by the three cells starting from L1 with the highest growth signal concentration in the template, see *Figure 3—figure supplement 1B*) and some cells were assigned a fixed concentration of 0 (a well-shaped arrangement of cells around the ones with a concentration different from zero, as can be seen in *Figure 3—figure supplement 1B*). In the model, this is done by running the process Model/Cell Ovule Growth/10 Growth Signal/a Set Cell Type/ Assign, after setting the variables to the desired values. The units have been a-dimensionalized.

Prior to pressurization, Dirichlet boundary conditions in the x-direction have been assigned to the lateral cell walls of the template (this means that those walls are not allowed to displace in the x-direction during pressurization). This condition has been assigned to simulate the presence of the surrounding lateral tissue. Dirichlet boundary conditions can be assigned by running the process Model/Cell Ovule Growth/20 Mass Spring Mechanics/a Set Dirichlet after having replaced the 3D null vector in the field 'Dirichlet' with a value different from zero for each coordinate to be fixed. This will affect all the selected nodes of the mass spring mesh.

The process Model/Cell Ovule Growth/20 Mass Spring Mechanics/b Set Cell Type Mechanics and Growth/Assign has been used to assign (i) the different growth directionality (see *Figure 3—figure supplement 1B*); (ii) different strain threshold for strain-based growth for the L1 cells and the inner ones with a growth signal different from 0 (the remaining cells deform elastically, but are not allowed to perform any growth) (see *Figure 3—figure supplement 1B*); (iii) different target area for cell division (53 µm$^2$ for L1 cells, 100 µm$^2$ for the SMC precursor, 80 µm$^2$ for the L2 cells neighbors of the pSMC, while all the other cells got the global target area for division of 63 µm$^2$, see *Figure 3—figure supplement 1B*). To correctly use the process, it is necessary to label L1 cells as 'L1' and the other cells with special assigned properties as 'Special Cell'.

Furthermore, the L1 is prescribed to perform only periclinal growth, by running the process Model/Cell Ovule Growth/20 Mass Spring Mechanics/d Only Periclinal Growth in L1 with the field 'Value' set to 'True'. When this option is assigned, the L1 cell edges corresponding to anticlinal walls are prevented from growing (this process will act on the cells previously labeled as 'L1').

It is now necessary to set the polarization field used to define growth directionality: this is obtained as the gradient of a diffusion process. The user needs to select cells with fixed high and low concentration (the specific value assigned here is arbitrary and functional only to be able to set a gradient field which will be normalized). In our simulation the cells in the left side of the template have been set with fixed high concentration, while the cells in the right side of the template have

been set with fixed low concentration (whether high/low concentration is fixed on the left or right corner is irrelevant): this produced the gradient visible in *Figure 3—figure supplement 1B*. To set the cells with fix concentration for the polarization process, the user needs to run the process Model/Cell Ovule Growth/30 Growth Process/a Set Fixed Concentr Diffusion Polarizer/Assign, after having assigned in the dialog box the desired values.

The remaining field values are global and defined in the Table of *Figure 3—figure supplement 1B*.

After this preparation, the template can be inflated to the prescribed pressure (in our model 0.5 Kg/s$^2$). It is important to stress out that in these simulations, the pressure adopted does not have the units of a physical pressure, but those of a force acting on a 1D segment (instead of on a 2D surface). Connected to this point, we stress out that these 2D simulations have a purely qualitative purpose and no quantitative comparison between the parameters used and the bio-physical quantities should be attempted.

After pressurization of the template, new Dirichlet boundary conditions are assigned: the nodes outlining the boundary of the template (except for the L1 top nodes) are blocked in all degrees of freedom (XYZ). This mesh is saved as starting point for the growth and cell division simulation with mass springs.

## 1.1.2 Confocal microscopy-based template preparation

A confocal stack from an almost flat placenta (dataset 2404_275a) has been used. The raw stack has been loaded in MorphoGraphX and by means of the clipping planes, a longitudinal central section of it has been selected. Then, the same procedure described in the MorphoGraphX manual (available here: https://www.mpipz.mpg.de/4085950/MGXUserManual.pdf) for generating $2\frac{1}{2}$D mesh has been followed, with the only difference that the surface mesh is generated with the CellMaker plugin as a rectangle with side length 0.5 $\mu$m. The signal projection depth has been calibrated so that only the walls of the midsection get projected (the longitudinal mid section has been as well selected accordingly to not display spurious walls from cells at different depth) and set to the value of 1 $\mu$. The final mesh gets converted into a 2D Cell Mesh by the process: Process/Mesh/Cell Mesh/Convert to 2D Cell Tissue with the field 'Max Wall Length' set to 10. After cleaning spurious boundary cells, and some manual curation to remove edges which are not junctions on the top wall of L1 cells (this to reduce the artifact of edge collapsing due to the absence of bending resistance during compressive forces), one obtains the final template to be used in the simulation within MorphoMechanX. The template preparation is then identical to what explained in the previous section and the distribution of cells with fixed high concentration is shown in *Figure 3—figure supplement 1F*.

## 1.1.3 Simulation cycle

The growth and division simulation starts from the pressurized template at mechanical equilibrium (see following). A simulation cycle consists of the following processes:

- growth signal diffusion calculation;
- polarization field calculation;
- mass spring mechanical equilibrium calculation;
- growth step;
- cell division.

The simulation cycle is repeated as many times as indicated by the simulation time T (which is not in physical units) mentioned in the main text, in *Figure 3* and *Figure 3—figure supplement 1*. We explain now in detail how each process is computed.

### Growth signal diffusion calculation

The growth signal is the result of a diffusive process acting on the cell level (on the dual graph). In this formulation, no sources or sinks are present: only some cells with a fixed concentration are specified at the beginning of the simulation. The discretized diffusion equation, following what done in *Bayer et al., 2009*, models the concentration variation inside a cell according to this formula (expressed in adimensionalized units through rescaling of the diffusion constant D):

$$\frac{d\rho_i}{d\tau} = \frac{D}{A_i} \sum_{j \in N_i} l_{ij}(\rho_j - \rho_i) \tag{1.1}$$

where $\rho_i$ indicates the signal concentration in the cell $i$, $\tau$, $D$ the diffusion coefficient, $A_i$ the area of cell $i$, $N_i$ is the set or cells connected to the cell $i$, while $l_{ij}$ is the length of the interface between the two cells. The rationale for this heuristic formula, formally identical to the discrete Laplace-Beltrami operator if the weight factors are made to coincide with $l_{ij}$ (**Reuter et al., 2009**), is that the variation of concentration in the cell $i$ depends on the difference of concentration between this cell and the neighboring ones rescaled by the cell surface (the bigger the cell, the less the concentration variation) and the length of the interface with each neighboring cell. Since we are looking for the steady state solution in the absence of sources and sinks, the rescaled diffusion coefficient $D$ does not affect the final solution and so it is set to one in the following calculations. To find the steady-state concentration, an adaptive forward Euler scheme was adopted (**Teukolsky et al., 1992**); the problem is easily solvable by a direct method, but this diffusion process has been designed to treat more general, non linear problems as well. So an iteration of the forward Euler scheme, to calculate the variation in concentration for the cell $i$ at the discrete simulation time step $n + 1$, is as follows:

$$\rho_i(\tau_{n+1}) - \rho_i(\tau_n) = \begin{cases} \frac{1}{A_i} \sum_{j \in N_i} l_{ij}(\rho_j(\tau_n) - \rho_i(\tau_n)) * \delta\tau, & \text{if } \rho_i \text{ is not fixed} \\ 0, & \text{otherwise} \end{cases} \tag{1.2}$$

The known fixed concentration are initialized with their assigned value. The time increment $\delta\tau$ is chosen based on an adaptive scheme which evaluates the magnitude of the concentration variation at the previous step to determine the new time increment (constrained between a preset max and min allowed time increment). The simulation is considered converged when the update on the concentration $\rho_i(\tau_{n+1}) - \rho_i(\tau_n)$ is smaller than a prescribed tolerance, more precisely when the mean between the averaging of the concentration update over all vertexes and the maximal of the concentration is below a threshold (concentrations taken as absolute values).

## Polarization field calculation

The polarization field is computed as the gradient of a diffusive process which is regulated by cells with a fixed concentration prescribed. The evolution process itself is computed as in the previous paragraph. Once the equilibrium concentration has been obtained, its 2D gradient (on the cell grid) is computed and then normalized. This gradient indicates the direction of the "polarizer' $\mathbf{n}_{Pol}$, which is used to control the direction of anisotropic growth. The advantage with respect to using fixed coordinates is that the polarization field is updated while the mesh changes, creating a system of local coordinates that follows the organ natural curvature while it grows (**Kennaway et al., 2011**).

## Mass spring mechanical equilibrium calculation

The mass-spring system is constituted by the network of linear springs which describe the cell walls in the template. The mechanical equilibrium is obtained when the internal reaction forces generated by the mass springs counterbalance the externally applied loads at each point of the mass spring network (MSN):

$$\mathbf{F}_{I_v} + \mathbf{F}_{E_v} = \mathbf{F}_{TOT_v} = 0 \quad \forall v \in MSN \tag{1.3}$$

where $\mathbf{F}_{I_v}$ indicates the internal reaction force at the node $v$, $\mathbf{F}_{E_v}$ the sum of the external forces applied to the node $v$ and $v$ belongs to the mass-spring-network ($MSN$). More specifically, the internal forces are given by the Hooke's law for mass-springs:

$$\mathbf{F}_{I_v} = k \sum_{\mathbf{u} \in N_v} (l_{u \to v} - \|\mathbf{p}_u - \mathbf{p}_v\|) \frac{\mathbf{p}_u - \mathbf{p}_v}{\|\mathbf{p}_u - \mathbf{p}_v\|} \tag{1.4}$$

$k$ is the spring constant (Kg/s$^2$), $N_v$ is the set of points which are connected to $v$, $l_{u \to v}$ is the rest length of spring connecting the point $u$ to the point $v$, while $\mathbf{p}_u$ and $\mathbf{p}_v$ are the respective positions.

Note that $\frac{\mathbf{p}_u - \mathbf{p}_v}{\|\mathbf{p}_u - \mathbf{p}_v\|}$ is the unit normal vector $\mathbf{n}_{u \to v}$ in the direction connecting the point $u$ to the point $v$. In our case forces are measured in $\mu$N and lengths in $\mu$m, but, being this a 2D representation of a 3D problem by means of 1D structures such as the mass springs, no quantitative comparison with bio-mechanical properties of the cell tissue should be attempted.

Regarding the externally applied forces, turgor pressure is considered. In our 2D mass-spring model this acts on the 1D line segment connecting two vertices in the mass spring system:

$$\mathbf{F}_{E_v} = \mathbf{F}_{P_v} = \frac{1}{2} \sum_{u \in N_v} \sum_{C \,|\, u \to v \in \mathcal{B}_C} P_C(l_{u \to v}) \mathbf{n}_{u \to v}^{\perp} \tag{1.5}$$

here $\mathbf{F}_{P_v}$ stands for the force due to turgor pressure on the node $v$, $\mathbf{n}_{u \to v}^{\perp}$ indicates the unit vector orthogonal to $\mathbf{n}_{u \to v}$ and pointing outwards with respect to the cell centroid. The summation is indented over all the vertexes $u$ connected to $v$ and over all the cells $C$ whose the edge $u \to v$ is part of the boundary ($\mathcal{B}_C$) and $P_C$ is the pressure afferent to cell $C$. To compute the forces equilibrium a semi-implicit Euler method has been adopted (*Teukolsky et al., 1992*):

$$\mathbf{p}(n+1) - \mathbf{p}(n) = h[\mathbf{1} - h\mathbf{J}(n)]^{-1} \mathbf{F}_{TOT}(n) \tag{1.6}$$

*Equation 1.6* is in a matrix form: $\mathbf{p}(n) = (\mathbf{p}_1(n), \cdots, \mathbf{p}_m(n))$, $\mathbf{F}_{TOT}(n) = (\mathbf{F}_{TOT1}(n), \cdots, \mathbf{F}_{TOTm}(n))$ and $\mathbf{J}(n)$ is the Jacobian (here computed numerically) of the force vector with respect to the nodal position vector $\mathbf{p}(n)$. $\mathbf{p}_i$ is the position of the $i$-th node in the *MSN*, while $n$ is the $n$-th iteration of the solver. The time stepping, $h$, is computed adaptively and based on the norm of the incremental displacement. The mechanical equilibrium is considered reached when the mean between the averaging of the norm of forces over all vertexes and the maximal force norm is below a threshold. The nodes with prescribed Dirichlet boundary condition will have the corresponding force equal to zero in the direction in which the condition is applied.

In case the option 'Wall Stiffness based on morphogen' in the process Model/Cell Ovule Growth/ 20 Mass Spring Mechanics/c Set Wall Stiffness/Assign has been set to true, for the selected cells during the simulation runtime, stiffness will be assigned within the range specified by the user in a linear negative correlation with respect to the growth signal.

## Growth step

After mechanical equilibrium has been reached, an incremental growth step, acting on the rest lengths of the mass springs ($l_{u \to v}$) is computed. Different types of growth can be specified and added together. In the model there are signal based growth and strain based growth, both can be explicitly required to occur along or orthogonal w.r.t. the polarization field direction $\mathbf{n}_{Pol}$. The incremental growth operator acting on the rest length of a mass spring segment connecting $u$ to $v$ is given as follows:

$$l_{u \to u}(T+1) = \delta \tau (G_{u \to v}^S + G_{u \to v}^{\epsilon} + G_{u \to v}^{\epsilon_{Pol}}) l_{u \to v}(T). \tag{1.7}$$

Here $G_{u \to v}^S$ is the signal based growth contribution, $G_{u \to v}^{\epsilon}$ is the strain based growth contribution, $G_{u \to v}^{\epsilon_{Pol}}$ is the strain based growth contribution in the polarizer basis ($\mathbf{n}_{Pol}, \mathbf{n}_{Pol}^{orth}$), $T$ is the discrete simulation time (number of simulation loop iterations) and $\delta \tau$ is a coefficient interpretable as a global growth scaling factor. More specifically:

$$G_{u \to v}^S = \begin{cases} K_{Par}^S & \text{if } abs(\mathbf{n}_{u \to v} \mathbf{n}_{Pol}) \geq 0.6 \\ K_{Per}^S & \text{if } abs(\mathbf{n}_{u \to v} \mathbf{n}_{Pol}) \leq 0.4 \\ 0.5(K_{Par}^S + K_{Per}^S) & \text{otherwise} \end{cases} \tag{1.8}$$

where $K_{Par}^S$ is the growth coefficient in the direction parallel to the polarization field, while $K_{Per}^S$ is the growth coefficient in the orthogonal direction. Since growth is performed by updating a rest length, the growth operator per-se can not affect directionality and introduce distortion. Therefore the notion of anisotropic growth is implemented by multiplying a rest length $l_{u \to v}(T)$ by $K_{Par}^S$ if its corresponding vector in the current configuration $\mathbf{n}_{u \to v}$ has a normalized projection along $\mathbf{n}_{Pol}$ bigger than 0.6. Vice-versa, if the projection is less than 0.4, the rest length will be multiplied by $K_{Per}^S$. For the

remaining cases, an average of the two coefficients will be adopted. Regarding strain-based growth we have:

$$G_{u \to v}^{\epsilon} = \begin{cases} K^{\epsilon}(\epsilon(T) - \epsilon_{thres}) & \text{if } (\epsilon(T) - \epsilon_{thres}) > 0 \\ 0 & \text{otherwise} \end{cases} \tag{1.9}$$

where $\epsilon(T) = (\|\mathbf{p}_u(T) - \mathbf{p}_v(T)\| - l_{u \to v}(T))/l_{u \to v}(T)$ is the linear strain computed at mechanical equilibrium at simulation loop time $T$ and $\epsilon_{thres}$ is the strain threshold. The rest length is not allowed to get bigger than the spring length in the current configuration.

The polarized strain based growth is prescribed as follows:

$$G_{u \to v}^{\epsilon_{Pol}} = \begin{cases} K_{Par}^{\epsilon}(\epsilon(T) - \epsilon_{thres}) & \text{if } (\epsilon(T) - \epsilon_{thres}) > 0 \text{ and } (\mathbf{n}_{u \to v}\mathbf{n}_{Pol}) \geq 0.6 \\ K_{Per}^{\epsilon}(\epsilon(T) - \epsilon_{thres}) & \text{if } (\epsilon(T) - \epsilon_{thres}) > 0 \text{ and } (\mathbf{n}_{u \to v}\mathbf{n}_{Pol}) \leq 0.4 \\ 0.5(K_{Par}^{\epsilon} + K_{Per}^{\epsilon}) & \text{otherwise} \end{cases} \tag{1.10}$$

The growth coefficients used in these formulas are specified cell-based, while growth is performed edge-based: therefore the coefficients actually used are an average quantity over the cells related to the edge.

## Cell division

Cell division is performed according to the rule 'shortest wall through the centroid' as described in *Mosca et al., 2018*. The target cell areas for division are assigned as described in Sec. 1.1.1. Cell wall pinching has been globally set to 0.1 (which corresponds to a proportionality factor connected to the new wall segment length or the distance between the newly inserted point and the cell centroid, depending on which is smaller), and is constrained to be overall smaller than 0.1 µm. The minimal distance from a pre-existing wall has been set to 0.8 µm. Cell wall sampling to search for the shortest wall has been set to 0.05 µm (the starting point for the sampling is aleatoric and in principle not guaranteed to be the same at different runs).

Upon cell division, cell properties (such as fixed growth signal concentration, fixed polarizer, cell type, target division area, special pressure, growth coefficients, etc.) are inherited by the daughter cells. For what concerns fixed high growth signal concentration, selective rules have been implemented to allow it to be propagated along a vertical line in the inner tissue. For what concerns cells in the inner layers, if a dividing wall is in the anticlinal direction, only one of the two daughter cells will inherit the fixed high signal concentration (the one with the smallest distance from the closest cell with a fixed high concentration). If the newly inserted wall is in the periclinal direction, if the mother cell (prior to division) is between two cells with a fixed high concentration, both daughter cells will inherit this property, otherwise only the one connected to the vertical stream of cells with a fixed high concentration. Regarding L1 cells, the daughter cells inherit both the fixed high concentration if the mother cell has two L1 neighbors with a fixed high concentration, otherwise only one cell will inherit this property with preference for the daughter cell with the shortest distance from the L2 cell with fixed high signal concentration. The propagation rules for fixed high concentration are not equivalent to a proper canalization model (which was not the aim of these simulations) and sometimes manual curation during the simulation time evolution has been required to ensure an uninterrupted vertical line of cells with fixed high concentration.

To assess the stability of the reference simulation (MS-Model 2), a robustness analysis has been performed by varying independently stiffness and growth signal intensity of 10% around the reference value. The results are qualitatively compatible with the reference model and available in the git repository: github.com/GabriellaMosca/MassSpring_2DovuleGrowthModel/tree/master/robustnessAnalysis.

## 1.2 Cell anisotropy quantification

Similarly to what done in *Louveaux et al., 2016*, a shape matrix has been defined for each cell. This matrix is the covariance matrix of the coordinates of the points along the cell outline. Given a distribution of points in the 2D plane, their covariance matrix with respect to their coordinates in the Cartesian plane is defined as

$$\mathbf{Cov} = \frac{1}{N_c} \begin{pmatrix} \sum_{i=0}^{N_c}(x_i - x_0)^2 & \sum_{i=0}^{N_c}(x_i - x_0)(y_i - y_0) \\ \sum_{i=0}^{N_c}(x_i - x_0)(y_i - y_0) & \sum_{i=0}^{N_c}(y_i - y_0)^2 \end{pmatrix} \tag{1.11}$$

where $x_i$, $y_i$ are the i-th point coordinates, $N_c$ is the number of points and $x_0$, $y_0$ are the arithmetic average of all the points coordinates.

This matrix is symmetric and so diagonalizable, its eigenvectors indicate the directions of maximal and minimal points dispersion (and so the anisotropicity of the distribution). More precisely, the eigenvector connected to the biggest eigenvalue defines the direction around which the standard deviation of the projection of the points position along that axis is maximal, while the other eigenvector defines the direction around which this same quantity is minimal. At the same time, one can see the first eigenvector as the one minimizing the variance of the distance of the points distribution from the direction it indicates, while the second eigenvector as the one maximizing instead this quantity.

If the points are uniformly distributed around a shape outline, the eigenvectors can be used as a proxy for the maximal and minimal elongation directions of the shape, in the sense described above. If one considers a rectangle (where the concept of shape anisotropy can intuitively be assimilated to the aspect ratio of its sides), the eigenvectors of this covariance matrix will be parallel to its sides and, if the sides ratio is greater than 1, also the corresponding eigenvalues will be greater than one. Furthermore the eigenvalues ratio depends on the sides lengths, but not on the rectangle area. If a rectangle has an aspect ratio greater than another (always computing longest size over shortest side), this relation will be valid also for the corresponding eigenvalues ratio.

If a rectangle is now a 2D cell which is growing and it is getting more elongated while not changing in the other direction, the eigenvalues ratio will increase. On the other hand if, while being anisotropic, it is still growing in an isotropic manner, its eigenvalues ratio will not change. So by looking at the eigenvalues ratio and their variation over time, it is possible to deduce the cell anisotropicity regardless of its surface and how it varies over time.

When a shape differs from that of a rectangle (or an ellipse), it is not possible to use the sides (main axes) aspect ratio as a measure of anisotropy, but the shape matrix provides an extension of this concept to other shapes. Clearly, the more the shape deviates form that of a rectangle, the more care should be used to interpret the results (especially with curved shapes, where a non-linear approach should be adopted).

In the measurements performed in this work, the cell perimeter has been enriched with intermediate points at an average distance prescribed by the user and the averaging has been weighted with the points distance to avoid artifacts due to non perfect homogenous points distribution. So the actual equation used to compute the covariance matrix is:

$$\mathbf{Cov} = \frac{1}{P_c} \begin{pmatrix} \sum_{i=0}^{N_c}(x_i - x_0)^2 l_i & \sum_{i=0}^{N_c}(x_i - x_0)(y_i - y_0)l_i \\ \sum_{i=0}^{N_c}(x_i - x_0)(y_i - y_0)l_i & \sum_{i=0}^{N_c}(y_i - y_0)^2 l_i \end{pmatrix} \tag{1.12}$$

Where $P_c$ stands for the cell perimeter and $l_i$ for the length of the perimeter segment containing the node $i$ in its middle. Furthermore, for the images shown in this work, it has been selected the option to measure cell anisotropy along the two orthogonal growth directions of the ovule. For this reason, by using $\mathbf{n}_{Pol}$ (the vector parallel to the polarization field direction, see Sec. 1.1.3) and $\mathbf{n}_{Pol}^{\perp}$, the orthogonal one, these two quantities are computed:

$$\alpha_{Pol} = \langle \mathbf{n}_{Pol}, \mathbf{Cov}\, \mathbf{n}_{Pol} \rangle, \qquad \alpha_{Pol}^{\perp} = \langle \mathbf{n}_{Pol}^{\perp}, \mathbf{Cov}\, \mathbf{n}_{Pol}^{\perp} \rangle \tag{1.13}$$

and their ratio is used as a proxy of cell anisotropy along the organ growth direction and the one orthogonal to it. More precisely, as in the figures the aim was to show the cell elongation along the ovule main growth direction, the anisotropy ratio displayed is given by this formula: $\alpha_{Pol}^{\perp}/\alpha_{Pol}$. This means that a displayed value of one indicates that the cell is fully isotropic, a displayed value bigger than one indicates that the cell is more elongated along the organ main elongation axis ($\alpha_{Pol}^{\perp}$), while a value between one and zero indicates that the cell is more elongated in the orthogonal direction to the main organ elongation axis. This calculation is performed in the model by running the process: Model/Cell Ovule Growth/50 Shape Quantifier/01 Compute Skew Symmetric Tensor, where the user

can specify the average distance for the virtual points to be placed along the shape perimeter for the computation (one should refine this number until the result is not significantly affected). The eigenvectors displayed in *Figure 3E* are scaled by their pertinent eigenvalue divided over the sum of all three eigenvalues (normalization), all globally rescaled by a length factor.

## 1.3 2D FEM based simulations for ovule growth - continuous limit

The finite element method (FEM) simulations are handled in MorphoMechanX by an in-house developed tool available at the git repository github.com/GabriellaMosca/FEM_2DOvuleGrowthModel (*Mosca, 2021c*). They run on a 2D mesh tessellated with triangles which constitute linear membrane elements with a mathematically prescribed thickness (not relevant in this type of simulation though).

### 1.3.1 Template preparation

The initial template is build as rectangle tessellated by squares assembled in 28 rows and 56 columns. The squares have a side of length 13.0527 µm (the value assigned to the length is arbitrary as these simulations aim at a qualitative analysis in 2D of a 3D problem). The tessellation squares have been each triangulated with four triangles sharing a vertex in the square center (see *Figure 3—figure supplement 1A*).

This template represents a portion of the ovule placenta as seen through a longitudinal cut. The first three lines or squares from the top are identified with the L1 layer and are assigned specific properties different from the remaining tissue, as will be described in the following.

The description which follows is referred to the reference case template, FEM-model 1, as from *Table 1* in the main text. The other models are obtained through minor modifications of the process described in the following.

As first thing, the growth signal intensity is assigned: this is obtained, similarly to what done with the mass springs, as the outcome of a diffusive process with some elements at a high and low fixed concentration. First of all, all the faces of the mesh need to be selected and the process Model/CCF/Fem Diffusion Growth/04 Create Diffusion Element has to be run, so that the element for growth signal diffusion is generated.

Then the nodes on the mesh with fixed signal concentration (high and low) need to be set. The nodes with high fixed growth signal concentration are forming a vertical line made of 16 rows by seven columns starting from the top margin of the mesh, in a centered position (see *Figure 3—figure supplement 1A* for the signal distribution). By running the process Model/CCF/Fem Diffusion Growth/03 Set Diffusion Dirichlet with the field 'Value' set to 0.2 (adimensional units adopted for diffusion as in the case of mass springs), the fixed high signal concentration is assigned to these vertices. Note that the 'Dirichlet Attribute' and 'Morphogen Attribute' (as well as the 'Element Attribute') names need to be assigned consistently for all the sub-processes contained in Model/CCF/Fem Diffusion Growth.

The vertices with a fixed low signal concentration are assigned with the same procedure. These vertices form the perimeter of a rectangle which encloses the vertical strip of cells with fixed high concentration. It is located at a distance from the mesh middle (in the horizontal coordinate) of 8 vertices, and from the mesh top (in the vertical coordinate) of 21 vertices (see *Figure 3—figure supplement 1A* for the distribution of the signal). The fixed assigned concentration is 0.

After assigning the initial conditions, the diffusion process itself is run by Model/CCF/Fem Diffusion Growth/00 Diffusion Solver. The details on how the diffusive process is handled are provided in the paragraph 1.3.2.

Similarly to what done for the growth signal, the material and growth main directions (E2, KPar) are obtained through a diffusive process (with the exception of the L1 layer, which is set with a different method). More precisely the gradient of the diffusive process will be used, or, alternatively, the direction orthogonal to it. The process for assigning the directions is Model/CCF/Fem Diffusion Anisotropy and is formally identical to Model/CCF/Fem Diffusion Growth.

It is possible to handle growth and material anisotropy with a unique diffusive process, where the source with fixed high signal concentration is located at the central vertex (in the horizontal direction) at the mesh top, while the vertexes with fixed low concentration are located at the mesh boundary (excluding the top boundary): the outcome of this diffusive process produces a radial gradient pointing in the direction of the vertex with fixed high concentration (see *Figure 3—figure*

*supplement 1A*, where the field is visible for the inner layers). For the mesh portion corresponding to the L1 layer, the material and growth anisotropy are set with a separate process, as it will be explained in the following.

At this point comes strictly mechanical part of the simulation. The mechanical elements need to be created by selecting all the faces of the mesh and by running the process Model/CCF/04 Reference Configuration which not only generates the elements for the FEM solver, but also sets their reference configuration as the current one. This process assigns also a thickness value, but this parameter is irrelevant in this 2D membrane simulation (so its has been left to 1 µm, as the membrane thickness won't affect the equilibrium configuration or the stresses, but will create a global rescaling factor for resultant forces).

Subsequently material properties get assigned. The template has a linear, hyperelastic, transverse isotropic, St. Venant material law, characterised by a special fiber direction (in this text named E2); for a detailed description of this material law and how it is used to compute the force vector for mechanical equilibrium (*Hofhuis et al., 2016*).

As first thing, it is necessary to assign the intensity of the Young modulus in the fiber direction (E2) and in the isotropic plane (denominated E1E3 in the text: in the template this is represented by a vector, but conceptually it is a plane because of the virtual thickness). The process Model/CCF/08a Set Aniso Dir sets global values for the Young modulus and Poisson ratio. For the described template, it needs to be run with the field 'Young E1E3' equal to '100', 'E2' to '500' and 'Poisson' equal to '0.3'. An additional stiffness, negatively proportional to the growth signal previously computed through diffusion, is assigned to the E1E3 plane by running Model/CCF/06b Material Properties based on Morphogens (for faces) in all the models as it creates a sharper delimitation of the ovule protrusion w.r.t to the placenta (for a comparison see the outcomes at the git repository github.com/GabriellaMosca/FEM_2DOvuleGrowthModel/tree/master/constantStiffness)

$$Y_{E1E3} + = \gamma \left( \frac{\text{MAX growth signal } - \text{ growth signal}}{\text{MAX growth signal } - \text{ MIN growth signal}} \right) \tag{1.14}$$

where $Y_{E1E3}$ is the Young modulus in the isotropic plane, $\gamma$ stands for the proportionality factor (set to 400), and growth signal is the signal obtained from the diffusive process Model/CCF/Fem Diffusion Growth(make sure to put the proper field signal name in the dialog box, its name was specified in Model/CCF/Fem Diffusion Anisotropy/05 Morphogen Visualize). The Young modulus distribution is the one shown in *Figure 3—figure supplement 1A*. In the mass springs simulation the stiffness was instead assigned constant in the template, as a replication of what done with the FEM simulation produced a numerically unstable model (see github.com/GabriellaMosca/MassSpring_2DovuleGrowthModel/tree/master/variableStiffness). We argue that for mass spring models, the effect of softer cell wall is already implicitly encoded in the combination of signal based growth and passive strain based growth, so that adding such an effect explicitly exaggerates the dishomogeneity between faster and more slowly growing cells. In the FEM simulation, lacking cellular resolution, intead, the passive strain based growth is mostly acting as a relaxation to accumulation of stresses due to tissue conflicts.

For the all the template, except for the L1 layer, the special fiber direction (E2) is given by the vector orthogonal to the gradient of the diffusive field computed in Model/CCF/Fem Diffusion Anisotropy. To assign it this way, after selecting the face elements of the template, the user needs to run Model/CCF/08b Set Aniso Dir Morphogens, where, in the field 'Direction Type' the key 'E2' has been assigned, in the field 'Direction from Diffusion Gradient' the key 'Orthogonal to Gradient' and in the field 'Morphogen Signal' the name chosen for the morphogenetic signal in Model/CCF/Fem Diffusion Anisotropy.

The L1 layer gets assigned a global E2 direction co-aligned with the y-axis (the vertical coordinate) by running the process Model/CCF/08a Set Aniso Dir with the accuracy of specifying direction type as 'E2' and 'Direction' as '0 1 0'.

The same procedure is repeated to assign the growth anisotropy direction (*Mosca et al., 2018*), denominated KPar in this text, with the difference that the field 'Direction Type' is 'KPAR' and the 'Direction from Diffusion Gradient' is 'Parallel to Gradient'. Regarding L1 layer, this is assigned a 'Direction' of '1 0 0'.

An edge-pressure acting on the template boundaries has been assigned (as the lateral and bottom boundaries will be constrained by Dirichlet condition, it is actually assigned only to its top boundary). This edge pressure acts exactly like the one described for the mass springs in *Equation 1.5* and has the same units. It has been set to a value of 5 Kg/s$^2$ and its role is to simulate, to some extent, the action of turgor pressure into generating stresses in the tissue.

Dirichlet boundary conditions (as with mass springs) are assigned on the bottom and lateral boundary of the template by running the process Model/CCF/15 Set Dirichlet and blocking all the degrees of freedom (setting the field 'Dirichlet Label' to '1 1 1').

As last, it is required to assign the growth field. In the reference simulation (FEM-model 1) signal based growth is purely in the polariser direction (KPar) combined with strain-based growth (see following for explanation). This is assigned for the selected faces by running the process Model/CCF/11b Set Growth Morphogens (on Faces) after setting the field 'KPar Scaling Factor' to 1, all the others to 1, and properly assigning the name in the field 'Morphogen Signal 1'.

## 1.3.2 Diffusion simulation

Diffusion simulation is used in this model to generate a concentration field given some fixed concentration nodes. The problem is solved in 2D on a flat surface. It is slightly different from the one implemented for mass springs (see paragraph 1.1.3) as it is based on the FEM and starts from a continuous formulation originally expressed by this equation (heat equation):

$$D\nabla c(\mathbf{x}) = 0 \qquad (1.15)$$

where $\nabla$ is the Laplace operator, $c$ the concentration of the diffusible substance and $\mathbf{x}$ are the coordinates used to described the space where diffusion is occurring. This problem requires boundary conditions (otherwise its solution would be trivial). For this diffusive process, unless prescribed Dirichlet condition (fixed concentration of the diffusing substance) have been assigned, zero flux condition is naturally assumed along the boundary. Internally to the domain, fixed concentration can be assigned as well, even if mathematically, this is equivalent to create an internal boundary (i.e. if inside a connected 2D surface a circle gets prescribed fixed concentration, which does not need to be constant in space, what will affect the solution outside this circle is only its boundary and not what has been set inside).

An explanation or introduction to the FEM is outside the purpose of these supplementary information, but there are several sources and manuals to consult, as for example *Hughes, 2012*. In this case a direct solver method has been adopted, which means that the problem can be written, after the FEM discretisation, in the following way:

$$\mathbf{Jc} = \mathbf{B} \qquad (1.16)$$

where $\mathbf{J}$ is a matrix, $\mathbf{c}$ is the vector of the nodal concentrations and $\mathbf{B}$ is a vector. The matrix $\mathbf{J}$, is given, in components, is as follows:

$$J_{ij} = \begin{cases} \sum_{\mathfrak{T} \ni i} \sum_{j \in N_i} A_{\mathfrak{T}} \left( \mathbf{D}\phi_i \mathbf{D}\phi_j^T \right) & \text{if } c_i \text{ not fixed by Dirichlet} \\ 1 & \text{else if } i = j \\ 0 & \text{otherwise} \end{cases} \qquad (1.17)$$

where the external sum is over all the triangles $\mathfrak{T}$ of the FEM mesh containing the node $i$, while the internal sum is over all the nodes $j$ which are connected to the node $i$. $A_{\mathfrak{T}}$ is the triangle area, $\mathbf{D}\phi_i$ is the vector of derivatives (in 2D) of the triangle linear basis functions $\phi_i$ (*Hughes, 2012*) referred to the triangle $\mathfrak{T}$ (which has been omitted for simplicity of notation) and node $i$, explicitly:

$$\mathbf{D}\phi_i = \begin{pmatrix} \partial \phi_i / \partial X \\ \partial \phi_i / \partial Y \end{pmatrix} \qquad (1.18)$$

with $X$ and $Y$ the local planar coordinates of the triangle. Regarding the B vector, it can be computed in components as follows:

$$B_i = \begin{cases} -\sum_{\mathfrak{F} \ni i} \sum_{j \in N_i} A_{\mathfrak{F}} c_j \left( \mathbf{D}\phi_i \mathbf{D}\phi_j^T \right) & \text{if } c_i \text{ not fixed by Dirichlet and } c_j \text{ fixed by Dirichlet} \\ 0 & \text{otherwise} \end{cases} \tag{1.19}$$

the final nodal concentration vector $\mathbf{c} = (c_1, \cdots, c_m)$ is obtained as the solution to the matrix equation $\mathbf{Jc} = \mathbf{B}$ after adding to it the fixed nodal concentration values.

In this calculation the diffusion coefficient $D$ has been omitted as it does not affect the steady state concentration in this type of diffusion.

### 1.3.3 One simulation cycle

A simulation cycle, which gets repeated in loop as many times (T) as specified by the user, consists of the following units:

- mechanical equilibrium simulation;
- growth step;
- mesh subdivision.

The template gets altered from a mechanical equilibrium condition because of the action of the edge pressure (assigned to the top margin) and of growth.

#### Mechanical equilibrium simulation

After pressurization and each growth step, the system generally needs to find a new equilibrium configuration which accommodates the pressure induced stresses as well as the residual ones introduced by the growth process (*Rodriguez et al., 1994*; *Goriely and Ben Amar, 2007* for an explanation of residual stresses).

As already introduced in Sec. 1.3.1, the material is transversely isotropic with a Sain-Venant Kirchhoff material law.

The equilibrium is computed by minimizing its strain energy by means of a semi-implicit Euler scheme (see *Equation 1.6*). The elemental contribution to the nodal-force vector calculation is provided in detail in *Hofhuis et al., 2016*. The final nodal-force vector is obtained by summing for each node all the elemental contributions it is connected to. The Jacobian of the force vector is obtained through numerical derivation.

Details of material parameters and numerical convergence are provided in the Table in *Figure 3—figure supplement 1A*.

#### Growth step

Growth is performed as a series of small deformations which are applied locally to the reference configuration and deform it without, directly, inducing stress or strain. As the deformation is local, no compatibility requirement is naturally satisfied and this has to be enforced by combining the growth deformation (which does not cause strain or stress) with an elastic deformation (which might induce strain and stress even in the absence of external loads), see *Rodriguez et al., 1994*.

The underlying hypothesis in this approach is that the timescale of growth is much bigger than that of elastic waves propagation, so that growth is supposed to occur in a quasi-static condition of mechanical equilibrium (*Goriely and Ben Amar, 2007*).

In this formulation it is assumed (with no loss of generality) that all the rotational components of the growth tensor are absorbed in the elasticity tensor, so that growth is described purely by a stretch tensor, with no rotational component. In this description of growth, for each growth step, the reference configuration is the one obtained at the previous growth step and not the original one and it is called virtual as it might not ever occur physically (because of incompatibility). So, if $\mathbf{F}_E^n$ and $\mathbf{F}_G^n$ are the elastic deformation tensor and the growth deformation tensor at the $n$-th iteration, if $\mathbf{X}_{n-1}$ is the virtual reference configuration obtained at growth step $n-1$, the global deformation $\mathbf{F}_{EG}$ at the $n$-th growth step will be as follows:

$$F_{EG}^n = \mathbf{F}_E^n \mathbf{F}_G^n \mathbf{X}_{n-1} \tag{1.20}$$

In general the growth tensor $\mathbf{F}_G^n$ might depend on the current body coordinates $\mathbf{x}_{n-1}$ or on its

state of elastic deformation/stress as computed from the previous virtual reference configuration. In the simulations in this paper, growth is constituted by an anisotropic part explicitly specified by the user through two diffusive process (one for the anisotropy direction $\mathbf{v}_{Kpar}$ and one for the growth intensity $KPar$, as described in Sec. 1.3.1) and a strain-based growth.

Growth is performed locally, by updating the rest length of the elemental triangles of the reference (possibly incompatible) configuration. It is possible to update this approach because triangles are simplexes, so their shape is fully determined by their side-lengths. The physical position of the reference triangle in space is not relevant.

Each triangle in the reference configuration stores in a ordered way the rest lengths of its edges and the angles made by the material fibers and growth anisotropy direction with its first side (the sides ordering is consistently preserved during the simulation). At the very first growth step, the growth direction and material anisotropy direction are naturally expressed in the reference configuration, so it is just a matter of computing the angle they make locally with the first triangle side. The growth anisotropy angle will be denominated $\alpha_{KPar}$, while the material anisotropy angle $\alpha_m$. The reference triangle is initially built from its ordered rest lengths in a local frame, where its first side is co-aligned and positively oriented with the x-axis. In this way

$$\mathbf{v}_{Kpar} = (cos(\alpha_{KPar}), sin(\alpha_{KPar})) \tag{1.21}$$

The growth tensor for this growth component, $\mathbf{G}_{KPar}$, is expressed in a diagonal basis:

$$\mathbf{G}_{KPar} = \begin{pmatrix} 0 & 0 \\ 0 & g_{KPar} \end{pmatrix} \tag{1.22}$$

where $g_{Kpar}$ is the growth coefficients for the local anisotropy growth in the direction parallel to $\mathbf{v}_{Kpar}$. In general it is possible to specify growth also in the direction orthogonal to $v_{KPar}$, but, depending on the growth hypotheses made, this direction might be orthogonal to $v_{KPar}$ in the current configuration, but not in the reference one. Or it might lose the notion of orthogonality in both configurations if it is assumed that growth fields are specified at the beginning of the simulations and get deformed during growth and elastic deformation. This of course, only in the case anisotropic growth is coupled with another growth tensor which makes the final growth operator non-diagonal w.r.t to $v_{KPar}$ anymore. It will not be exposed here how the growth tensor should be computed in these cases.

Consequently, it is necessary to rotate the reference triangle so that $\mathbf{v}_{Kpar}$ is representable in a canonical basis coordinates: given the explicit form of $\mathbf{G}_{KPar}$, as (0, 1).

The strain-based growth tensor $\mathbf{G}_E$, which will be summed up with the purely anisotropic one ($\mathbf{G}_{KPar}$) to constitute a global growth tensor $\mathbf{G}_{TOT}$, is expressed in the diagonal basis of $\mathbf{G}_{KPar}$. The strain here considered is the Green-Lagrange strain tensor, which is expressed in the current body configuration (more specifically it maps a vector in the reference configuration to a vector in the current one). The computation of $\mathbf{G}_E$ components proceeds as follows:

- the triangle represented in the current 3D configuration gets mapped by a rotation matrix in 2D;
- the Green-Lagrange strain tensor between the current triangle (mapped in 2D) and the reference one is computed (see *Hofhuis et al., 2016*) and eigenvalues and eigenvectors are derived;
- after computation of the engineering strain for each in-plane eigenvalue, the strain-based growth matrix in diagonal representation is built:

$$\mathbf{G}_E^D = g_\epsilon \begin{pmatrix} max(0, \epsilon_{11}^E - \epsilon_0^E) & 0 \\ 0 & max(0, \epsilon_{22}^E - \epsilon_0^E) \end{pmatrix} \tag{1.23}$$

- where $\epsilon_{11}^E$ and $\epsilon_{22}^E$ are the ordered engineering strain eigenvalues obtained from the Green-Lagrange strain eigenvalues $\epsilon_{11}$ and $\epsilon_{22}$ ($\epsilon_{ii}^E = -1 + \sqrt{(1 + 2\epsilon_{ii})}$), while $\epsilon_0^E$ is a threshold value below which no strain based growth occurs and $g_\epsilon$ is a growth proportionality factor (for each principal direction; the engineering strain has been chosen due to its linear dependence on the stretch);
- the strain-based growth tensor gets represented in the same basis which diagonalizes $\mathbf{G}_{KPar}$:

$$\mathbf{G}_E = \mathbf{R}^T_{eigen} \mathbf{G}^D_E \mathbf{R}_{eigen} \qquad (1.24)$$

- where $\mathbf{R}^T_{eigen}$ is the matrix formed by the Green-Lagrange strain eigenvectors as columns.

The final growth tensor is then:

$$\mathbf{G}_{TOT} = \mathbf{1} + dt(\mathbf{G}_{KPar} + \mathbf{G}_E) \qquad (1.25)$$

and it is expressed in the basis which diagonalises $\mathbf{v}_{Kpar}$. By applying it to the reference triangle coordinates, the triangles is grown (at the end only the rest lengths are stored, as the triangle arrangement in space is irrelevant).

It is now necessary to update the growth anisotropy and material angle with respect to the grown triangle first side, $\alpha_{KPar}$ and $\alpha_m$ respectively. The reference triangle configuration has been rotated so to be in the diagonal basis for $\mathbf{G}_{KPar}$, so the same rotation has to e applied to $\mathbf{v}_m$:

$$\mathbf{v}_m = \mathbf{R}_{KPar}(cos(\alpha_m), sin(\alpha_m)) \qquad (1.26)$$

where by definition $\mathbf{R}_{KPar}$ is the rotation matrix which rotates $\mathbf{v}_{KPar}$ into the vector represented by the coordinates (0, 1). At this point, by applying the final growth operator $\mathbf{G}_{TOT}$ to the two vectors, one obtains their orientation (plus an irrelevant stretch) in the new grown configuration, so it is possible to compute the updated angles between these two vectors and the first, consistently oriented, triangle side.

Underlying this approach there is the idea that the anisotropy direction got computed at the beginning of the simulation and gets passively deformed by the growing rest configuration. An alternative approach would have been to re-compute the diffusion process and its gradient at each growth step on the current configuration and then map it on the rest configuration. In this specific simulation scenario, it is possible to verify that the two gradients so obtained do not differ significantly.

## Mesh subdivision

When a triangle area in the current configuration exceeds a pre-set threshold (Max Triangle Area in Table in *Figure 3—figure supplement 1A*), it gets divided by a bisection algorithm which splits its longest side in two segments of equal length and generates so two new triangles (which replace the old one) by connecting the newly inserted vertex to the opposed one. To ensure a conformal mesh, the subdivision is propagated to neighbor triangles as described in *Rivara and Inostroza, 1995*. After each subdivision the triangle, edge and vertex properties need to be propagated

- the vertex properties which need to be propagated to the newly inserted vertex are the Dirichlet condition (both for mechanics and diffusion): if the new vertex has been inserted between two vertexes with Dirichlet conditions assigned, it will inherit them, otherwise no;
- an edge in the current configuration gets divided in two equal parts, so the new edges inherit half the rest length of the undivided one each;
- by splitting an edge, the initial triangle is split in two smaller triangles (not necessarily identical): each of them inherit the same triangle properties of the undivided one (i.e. material properties, growth properties, anisotropy angles, etc.).

## Robustness analysis

As done with mass springs, a robustness analysis around the reference model (FEM-Model 1) has been performed by varying independently stiffness and intensity of growth signal of 10% around the reference values. As can be observed by examining the results in the git repository github.com/GabriellaMosca/FEM_2DOvuleGrowthModel/tree/master/robustnessAnalysis such variations do not affect significantly the outcome, confirming the stability of the reference model.

### 1.4 3D analysis of ovule morphology and connectivity

#### 1.4.1 Ovule segmentation export from IMARIS

A in-house tool (ExportImarisCells, git repo github.com/barouxlab/ExportImarisCells, *Mosca, 2021a*) has been developed to export segmentations performed in the proprietary software IMARIS (imaris.oxinst.com). This tool, to be launch within IMARIS, creates a cell-mask with a different integer value for each segmented cell and the value zero for everything else. All the cell-masks are summed algebraically together to provide a final mask where each cell (each 3D voxel occupied by a cell) is labeled with a different integer value and the outside space is labeled with the value 0. This algorithm is based on the hypothesis that cells occupy mutually exclusive space regions. It is possible to export with different labels $2^{16}$ different cells. The output of this operation is saved as a tiff images series.

#### 1.4.2 3D mesh cell-based and surface mesh creation in MorphoGraphX

After exporting the segmentation as a tiff-file series from IMARIS in the previous step, it is now possible to load it in MorphoGraphX (*Barbier de Reuille et al., 2015*, https://morphographx.org/). The user needs to specify manually the z voxel size. At this point the stack is loaded and visible in MorphoGraphX and the option 16 bit as well as label need to be selected to proceed properly. Upon running the process Process/Stack/Segmentation/Relabel the image series will be relabeled and can be saved as a mgxs file, a native format in MorphoGraphX for segmented stacks. After this step, it is possible to properly generate the cell mesh by running the process Process/Mesh/Creation/Marching Cubes 3D, where the cube size, which is related to the mesh refinement, has to be specified by the user (a size of 0.5 $\mu^2$ has been specified for meshes connected to ovules till Stage 1-I, for later stages, to avoid underestimation of surface of contact between cells, a size of 0.1 $\mu^2$ has been adopted). This provides a cell mesh which can already be used for volume quantifications. As a further step, to be able to use it in MorphoMechanX, the cell mesh needs to be converted and saved as a 3D tissue by running the process: Process/Mesh/Cell Mesh/Convert to 3D Cell Tissue and then saving the result. Both mesh types can be saved in the native MorphographX mesh format mgxm. Starting back from the saved relabeled stack (the mgxs file), by following the instructions provided in the MorphoGraphX user manual (available here https://www.mpipz.mpg.de/4085950/MGXUser-Manual.pdf) it is possible to generate a surface mesh and by running Process/Mesh/Signal/Project Mesh Curvature the curvature signal will be projected on it. The curvature used in the analysis reported in *Figure 1E–F* is the minimal curvature, with manually set max and min range (for Stages 0-I and 0-II the range has been set between −0.15 and 0.15, while for later stages, as the ovule profile gets sharper, between −0.25 and 0.25).

#### 1.4.3 Connectivity analysis for the ovule layers

The 3D Cell Tissue mesh saved as described in the previous section can be loaded in MorphoMechanX to perform a semi-automated cell layer classification and connectivity analysis. An in-house tool to be run within MorphoMechanX has been developed to be able to detect semi-authomatically cell layers, after the user selects manually the L1 cells. If the user selects as well explicitly the pSMC/MMC, this will be marked as so in the analysis ad the companion cells will be identified (both cell types are considered sub-cases on L2 layer cell classification). The tool is available at the git repository: github.com/GabriellaMosca/3DAutoLabeling-ShapeQuant (*Mosca, 2021d*).

A cell is identified to be a L2 cells if it shares a portion of its wall with the L1 layer. To avoid spurious contacts due to segmentation imperfection, a minimal threshold for contact can be specified (in this case 0.01 $\mu^2$). Once the L2 cells have been identified (the pSMC/MMC and companion cells are considered L2 cells), the L3 cells will be detected as the cells sharing a portion of their wall with L2, but none with L1. Finally all other cells are gathered in a global group. This tool will save to a file, for each selected cell, its labeling (L1, L2, pSMC/MMC, L3, etc.), its volume and in case of L2 cell (including pSMC/MMC cell) its surface of contact with L3 layer. In the file are furthermore saved the three eigenvalues of the positional covariance matrix described in the next section. For a tutorial on how to use the tool, see the README.md file in the git repository.

## 1.4.4 MMC shape characterization with its positional covariance matrix

Following the idea presented in Sec. 1.2 of this document, a 3D version of the covariance matrix of *Equation 1.11* has been used to characterize anisotropy of pSMC/MMCcell shape in 3D. The computation, analogously with what done for the 2D case, is performed along the cell surface and the already existing mesh is used for the point distribution. The points used for the computation are the centroids of each mesh triangle and the covariance matrix, with elements area as weights to compensate for dis-homogeneities in the mesh looks like:

$$
\mathbf{Cov} = \frac{1}{\sum_i^N A_i} \begin{pmatrix} \sum_{i=0}^{N} A_i(x_i - x_0)^2 & \sum_{i=0}^{N} A_i(x_i - x_0)(y_i - y_0) & \sum_{i=0}^{N} A_i(x_i - x_0)(z_i - z_0) \\ \sum_{i=0}^{N} A_i(x_i - x_0)(y_i - y_0) & \sum_{i=0}^{N} A_i(y_i - y_0)^2 & \sum_{i=0}^{N} A_i(y_i - y_0)(z_i - z_0) \\ \sum_{i=0}^{N} A_i(x_i - x_0)(z_i - z_0) & \sum_{i=0}^{N} A_i(y_i - y_0)(z_i - z_0) & \sum_{i=0}^{N} A_i(z_i - z_0)^2 \end{pmatrix}
\tag{1.27}
$$

in this formula $N$ is the total element number (not the number of nodes), $A_i$ is the element area, $x_i, y_i, z_i$ are the elements centroids and $x_0, y_0, z_0$ are the overall 3D shape centroid (computed as the weighted average of the element centroids positions).

In this case the three eigenvectors of the covariance matrix correspond to (i)the axis where the variance of the points projection on it is maximal, (ii) the axis where it is minimal and (iii) a third orthogonal direction to the other two where the variance is intermediate.

To get a proxy of cell anisotropy (so independent from the cell size) it is possible to rescale the eigenvalues of the covariance matrix by dividing each for the sum of all three (making them a partition of unity):

$$
c_i^{rescaled} = \frac{c_i}{\sum_{j=1}^{3} c_j}
\tag{1.28}
$$

where $c_i$ are the eigenvalues of the covariance matrix. Furthermore, to be able to compare the principal anisotropy directions of the analyzed cells, the eigenvalues were saved in a rigid Cartesian frame co-aligned with the ovule main elongation axis.

