## [Decision Letter]

**Acceptance summary:**

The authors use imaging analysis of developing ovule primordia in Arabidopsis to clarify the importance of organ morphogenesis in cell fate. They document the growth of ovule cells in three dimensions, then use computational modelling to predict factors underlying ovule growth, shape and spore mother cell (SMC) differentiation. This elegant work adds new information and confirms previous hypotheses for the field.

**Decision letter after peer review:**

Thank you for submitting your article "Organ geometry channels reproductive cell fate in the Arabidopsis ovule primordium" for consideration by *eLife*. Your article has been reviewed by 2 peer reviewers, and the evaluation has been overseen by a Reviewing Editor and Jürgen Kleine-Vehn as the Senior Editor. The reviewers have opted to remain anonymous.

Essential revisions:

The manuscript is difficult to follow, i.e. it is necessary to read very meticulously through the text, figures and supplementary figures in order to understand the results. This is especially true for the results of the simulation experiments. To aid the reader, Table 2 should be integrated with all output images, e.g. Figure 3-suppl.1B. In the text, FEM-models were explained first, followed by explanation of MS-models. It would be better to explain both models in parallel in each simulation case.

Regarding the katatin mutant:

One concern is that the authors show effects on of kat on several cell types, but these vary between L1, L2 and SMC. It is essential to know where the katanin gene is expressed in the ovule to understand the impact on growth and SMC differentiation. This would ideally be provided via in situ hybridization or a fluorescent protein reporter.

Also in this section, the image for the DMC1-GUS marker is not convincing, given that GUS easily leaks to surrounding cells. The increased frequency of expression to 57% is also not consistent with the other fluorescent markers. Is this pattern explained by normally-sized L2 neighbors expressing DMC1? Do these cells enter meiosis? Further clarification is required.

The authors also analyzed WOX2-CenH3, a marker that in WT is expressed in the functional megaspore nucleus. They observed around 20% of mutant ovules expressing the marker in two cells, one being the functional megaspore and the other being an ectopic spore. While enough information is provided in the supplementary figure, in the main text it is not well explained that the ectopic spore results from the survival of an extra spore from the same tetrad, and not a result of ectopic cSMC now expressing this gametophytic marker. This needs to be clarified in the text, and adding a photograph of a tetrad stage for this marker might help clarify the abnormality observed.

Together, the DMC1 and the WOX2-CenH3 marker results appear to indicate that KAT may be part of another mechanism that involves spore selection/survival. While this information is not conveyed in the main text and perhaps is out of scope, is it possible that the kat mutation leads to an overall increase in germline identity in the diploid and haploid generations? In any case, it is difficult to clearly identify these cells expressing WOX2-CenH3 as spores, and it would be important to know if these cells continue gametophytic development or not.

The authors analyzed another cell cycle marker, PCNA1-GFP, in both WT and kat L2 cells. As written, it is difficult to understand what precisely occurs. Despite this, the authors conclude that even WT ovules with multiple cSMCs present S-phase pattern of the marker, and as canalization occurs, ovules present this pattern only in one SMC. Does this mean that multiple L2 cells can initiate replication of the DNA despite not showing mitotic activity (regarding the initial results obtained with Cyclin marker)? It seems that not all the ovules express the marker if, for example, at stage 0-II, 25% of ovules showed the S-phase pattern in one SMC and 30% of ovules in two cSMCs. The authors need to consider what happens in the other 45% of ovules.

For this marker, the difference in kat background is a higher percentage of ovules that still present multiple cells with S-phase pattern at stages where canalization should be occurring. The authors suggest that in the abnormal cells S-phase entry is altered and prolonged. This data fits well the remaining observations made through the work and adds the notion that the abnormal cells may somewhat be preparing for meiosis similarly to the main SMC, despite that appeared not to occur. Along these lines, why not report aniline blue staining patterns for the katanin mutant, as is the case in many multi-SMC studies? Callose is a hallmark of MMC (SMC) specification but does not accumulate in the multiple early SMCs in WT. Do the cells in the katanin mutant go so far as to accumulate this cell wall component?

L257-259The authors describe "SMC characteristic distinct size, anisotropic shape and orientation aligned with the growth axis of the primordium emerge already in Phase I". However, where is the figure supporting "SMC anisotropic shape" emerges in Phase I? Does it correspond to Figure 3C, E or Figure 3 suppl1 D? Make it clearer in the text.

L602-606 the descriptions about wild-type SMC candidates should be moved to p.11, and discussed together with anisotropic development of SMC through phases I and II. Descriptions on p. 26 should concentrate on mutant SMCs.

line 327 Should white be blue? – there is no white in the figure.

Why use "digit-shaped" to describe the ovule primordia. It is standard in the field to use "finger-like".

Line 47 insert the word and, e.g. should read intrinsic factors and signals from the surrounding".

Line 57 Rephrase, i.e. Arabidopsis, maize and rice mutants in which SMC singleness…"

Line 81 clearing, not clearings.

Line 163 and 165. In one case 4.5 fold, in the next 50%. Please use one or the other.

line 197, what do the colors mean in 2B?

line 313 doesn't make sense, rephrase.

In OvuleViz, it would be nice to have some of the 3D segmentations there to use as template.

Please make it clear throughout the manuscript if WT or mutant ovules are being discussed. For example, line 634 to 638.

On page 7, perhaps would be more interesting to substitute table1 for a bar graph?

Figure 2—figure supplement 1D needs cell labels and color annotations.

Figure 3—figure supplement 1 is missing color scales for each of those models and what they represent.

Figure 6 F and H – essentially the same analysis done in two different phases, but shown differently, in one case grouped by class A or B ovules and the other by SMC or neighbors. Consider making them more similar.

---

## [Author Response]

Essential revisions:The manuscript is difficult to follow, i.e. it is necessary to read very meticulously through the text, figures and supplementary figures in order to understand the results. This is especially true for the results of the simulation experiments. To aid the reader, Table 2 should be integrated with all output images, e.g. Figure 3-suppl.1B. In the text, FEM-models were explained first, followed by explanation of MS-models. It would be better to explain both models in parallel in each simulation case.

We revised the text, notably the section describing the simulation work, following the reviewers’ suggestions. In addition, we provide a schematic representation of the main parameters of the models and their variation in Figure 3 and Figure 3- supplement figure 1 to aid for the interpretation.

Regarding the katatin mutant:One concern is that the authors show effects on of kat on several cell types, but these vary between L1, L2 and SMC. It is essential to know where the katanin gene is expressed in the ovule to understand the impact on growth and SMC differentiation. This would ideally be provided via in situ hybridization or a fluorescent protein reporter.

We now provide the analysis of a functional GFP-tagged KATANIN reporter line (GFP:KTN1; *ktn1-2* Lindeboom et al. 2013 doi: 10.1126/science) in whole-mount, fresh ovules. This reporter line marks microtubule severing foci enriched in KATANIN. These foci are highly unstable within a few minutes following ovule mounting rendering high-resolution imaging very challenging. We provide here as Figure 4-supplement figure 1, a panel of representative images (of a total of 59 observed ovules) at different ovule primordia stages showing KATANIN foci at cell boundaries of all L1 cells and also in L2 cells, including the SMC candidate and its neighbors. This confirms the ubiquitous presence of KATANIN in the ovule primordium and supports the hypothesis that ectopic SMC candidates in the *katanin* mutant arise from a global effect (altered ovule geometry), rather than a cell-specific effect. Differences in KATANIN accumulation and/or dynamics between the layers are not excluded, however addressing this point would require functional domain-targeted analyses, beyond our scope here.

We also added two sentences in the text regarding KAT localization: in the result section (L451): “Note that *KAT* is expressed ubiquitously and, thus, also in the ovule: the KATANIN protein could be detected in both epidermal and internal L2,L3 tissue layers using a GFP reporter (Figure 4—figure supplement 1).”, and in the discussion (L756): "KATANIN is present throughout ovule primordium cells (Figure 4—figure supplement 1)”.

Also in this section, the image for the DMC1-GUS marker is not convincing, given that GUS easily leaks to surrounding cells. The increased frequency of expression to 57% is also not consistent with the other fluorescent markers. Is this pattern explained by normally-sized L2 neighbors expressing DMC1? Do these cells enter meiosis? Further clarification is required.

We agree that GUS easily leaks, however we used here highly stringent conditions, and observed few events of ectopic expression in wild-type, and never in L1 cells in *katanin* for instance, suggesting that the observed ectopic expression of pDMC1:GUS results are not due to diffusion of the enzymatic reaction product. The increased frequency as compared to other markers in *kat* indeed suggests that additional L2 neighbors are able to express DMC1, which is possibly required but not sufficient for meiotic completion.

To clarify if ectopic cells in *kat* are able to complete meiosis, we performed callose staining with aniline blue, as suggested by the reviewers below. The results, presented now in the text and in Figure 5—figure supplement 2B, suggest that no ectopic meiosis occurs in *kat*.

The authors also analyzed WOX2-CenH3, a marker that in WT is expressed in the functional megaspore nucleus. They observed around 20% of mutant ovules expressing the marker in two cells, one being the functional megaspore and the other being an ectopic spore. While enough information is provided in the supplementary figure, in the main text it is not well explained that the ectopic spore results from the survival of an extra spore from the same tetrad, and not a result of ectopic cSMC now expressing this gametophytic marker. This needs to be clarified in the text, and adding a photograph of a tetrad stage for this marker might help clarify the abnormality observed.Together, the DMC1 and the WOX2-CenH3 marker results appear to indicate that KAT may be part of another mechanism that involves spore selection/survival. While this information is not conveyed in the main text and perhaps is out of scope, is it possible that the kat mutation leads to an overall increase in germline identity in the diploid and haploid generations? In any case, it is difficult to clearly identify these cells expressing WOX2-CenH3 as spores, and it would be important to know if these cells continue gametophytic development or not.

We collectively address these comments by providing a detailed analysis of an additional gametophytic fate marker, pAKV:H2B-YFP (Schmidt et al., 2011) and an analysis using callose staining as a marker of the meiotic tetrad, as suggested by the reviewers. The results are presented in Figure 5—figure supplement 2B-D and commented in the main text. The analysis suggests that ectopic SMC are unlikely to resume meiosis, and instead may re-enter a somatic fate, and that ectopic expression of pWOX2:CENH3-GFP and pAKV:H2B-YFP likely arises from surviving spores not able to form ectopic embryo sacs.

The authors analyzed another cell cycle marker, PCNA1-GFP, in both WT and kat L2 cells. As written, it is difficult to understand what precisely occurs. Despite this, the authors conclude that even WT ovules with multiple cSMCs present S-phase pattern of the marker, and as canalization occurs, ovules present this pattern only in one SMC. Does this mean that multiple L2 cells can initiate replication of the DNA despite not showing mitotic activity (regarding the initial results obtained with Cyclin marker)? It seems that not all the ovules express the marker if, for example, at stage 0-II, 25% of ovules showed the S-phase pattern in one SMC and 30% of ovules in two cSMCs. The authors need to consider what happens in the other 45% of ovules.

Indeed, the results suggest that multiple L2 cells have the competence to initiate DNA replication despite having exited mitotic cell cycle as suggested by the CYCLIN marker analysis. The frequencies that the reviewer refers to correspond to a snapshot at a specific stage. The dynamic of this marker is not synchronous between all ovules. For the 45% ovules at stage 0-II not showing the PCNA speckled S-phase pattern, the S-phase may have occurred earlier or will initiate later, relative to the time of observation. Given the current complexity of data presentation as pointed by the reviewers, we did not feel necessary to comment in the text on this aspect which does not influence the major conclusion.

For this marker, the difference in kat background is a higher percentage of ovules that still present multiple cells with S-phase pattern at stages where canalization should be occurring. The authors suggest that in the abnormal cells S-phase entry is altered and prolonged. This data fits well the remaining observations made through the work and adds the notion that the abnormal cells may somewhat be preparing for meiosis similarly to the main SMC, despite that appeared not to occur. Along these lines, why not report aniline blue staining patterns for the katanin mutant, as is the case in many multi-SMC studies? Callose is a hallmark of MMC (SMC) specification but does not accumulate in the multiple early SMCs in WT. Do the cells in the katanin mutant go so far as to accumulate this cell wall component?

Thank you for the suggestion, we now provide an analysis using aniline blue staining Figure 5—figure supplement 2B. We did not find evidence that ectopic SMCs in *katanin* ovules accumulate callose similarly to what could be expected for normal SMC. However, a higher frequency of *katanin* ovules seem to fail accumulating callose in comparison to the wild-type, as shown in a new graph Figure 5—figure supplement 2B., suggesting cell wall defects in *kat,* which might also influence SMC fate.

L257-259The authors describe "SMC characteristic distinct size, anisotropic shape and orientation aligned with the growth axis of the primordium emerge already in Phase I". However, where is the figure supporting "SMC anisotropic shape" emerges in Phase I? Does it correspond to Figure 3C, E or Figure 3 suppl1 D? Make it clearer in the text.

This corresponds to the result described 3 lines above this sentence (“We measured the degree of alignment of the SMC …. (Figure 2K). This confirmed that the SMC has a consistent anisotropic shape from early stage onwards”). Since the sentence “SMC characteristic…” is part of a summary of this result section (“taken together.”) we did not re-cite all figure panels associated with the conclusions we draw collectively from all observations presented Figure 2 and Figure 2- supplements.

L602-606 the descriptions about wild-type SMC candidates should be moved to p.11, and discussed together with anisotropic development of SMC through phases I and II. Descriptions on p. 26 should concentrate on mutant SMCs.

The concept of SMC Class A and B can only be introduced on the basis of a large sample, as multiple SMC candidates occur only at a frequency of 20% in the wild-type Phase I and were not tangible in the 3D image dataset. The reviewers’ suggestion of reorganization implies to move not only the description of the SMC candidates L602-606 but the entire section explaining the concept of progressive restriction of SMC fate. The phenotype of the mutant SMCs in *katanin* are described in the previous section, with Figure 5 and associated supplements. The current section “SMC singleness is progressively resolved during primordium growth” with Figure 6 and supplements addresses the impact of the *katanin* mutation on fate restriction during developmental progression. We feel that a reorganization as suggested would disrupt the flow by introducing ClassA-B ovules and progressive SMC fate restriction at the beginning of the manuscript and coming to it again two Results sections later, while not allowing a focused conceptual explanation. Hence, we suggest to keep the current structure for the sake of clarity.

line 327 Should white be blue? – there is no white in the figure.

The white color refers to the fine lines indicating the principal direction of stress, as classically done in the modeling field. The sentence has been modified for clarity “The level of accumulated strain is indicated by the background coloring (according to the heatmap) while the principal directions are shown as fine lines (white: expansive, red: compressive).”

Why use "digit-shaped" to describe the ovule primordia. It is standard in the field to use "finger-like".

In morphogenetics, the term ‘digit’ is common to refer to organ structures forming the finger or equivalent body part in animals (for instance ‘Digit patterning during limb development’).

Line 47 insert the word and, e.g. should read intrinsic factors and signals from the surrounding"

Corrected. “..by both intrinsic factors and signals from the surrounding somatic tissues”.

Line 57 Rephrase, i.e. Arabidopsis, maize and rice mutants in which SMC singleness…".

Corrected.

Line 81 clearing, not clearings.

Corrected.

Line 163 and 165. In one case 4.5 fold, in the next 50%. Please use one or the other.

We agree that it is principally better to use one comparative order. Yet 4.5 fold would read 450% and 50% would read half-a-fold. Neither sounded very elegant to us, this is why we opted for this formulation.

line 197, what do the colors mean in 2B?

It is indicated in the legend “Color scale: minimal curvature mm-1 (see also Figure 2—figure supplement 1B).”

line 313 doesn't make sense, rephrase.

The sentence was rephrased for “This produced a sharp primordium, narrower and taller than the primordium in the Reference Model, and a thicker L1 layer in the FEM model”.

In OvuleViz, it would be nice to have some of the 3D segmentations there to use as template.

There is already a template file called “Segmented_Dataset” available with the comment “Merged Dataset Example for Interactive Analysis”.

Please make it clear throughout the manuscript if WT or mutant ovules are being discussed. For example, line 634 to 638.

In this section there is no alternating commentaries on either wt or mutant ovules, rather, the results are commented separately into two parts starting with “In Phase I wild-type primordia,…” followed by “In *kat* primordia”. For clarity we introduced a line return at the beginning of each paragraph.

On page 7, perhaps would be more interesting to substitute table1 for a bar graph?

This suggestion would require several bar graphs for all the entries and generate an additional figure which we do not feel necessary. The measures in Table 1 do not bear a biological significance worth commenting extensively. Instead, the table rather aims at providing a pragmatic guide, with numbers that are concrete measures for classification of the stages in future studies.

Figure 2—figure supplement 1D needs cell labels and color annotations.

We added the sentence in the legend “ the color code follows that of the cell label in panel C”.

Figure 3—figure supplement 1 is missing color scales for each of those models and what they represent.

We revised the legend with the following sentence for panel B and C “ color scale: same as in panel A and D for FEM- and MS-models, respectively”.

Figure 6 F and H – essentially the same analysis done in two different phases, but shown differently, in one case grouped by class A or B ovules and the other by SMC or neighbors. Consider making them more similar.

Panels F and H do not exactly refer to the same analysis. F compares the event between class A and B for SMC and neighbors, whereas panel H compares the events between the SMC and neighbors between wild-type and mutant. For the later, we follow the color code (wt, mutant) adopted throughout the manuscript. We thus do not see the possibility to make them more similar without overloading the figure.